# MODE: Learning compositional representations of complex systems with Mixtures Of Dynamical Experts

## Abstract

Dynamical systems in the life sciences are often composed of complex mixtures of overlapping behavioral regimes. Cellular subpopulations may shift from cycling to equilibrium dynamics or branch towards different developmental fates. The transitions between these regimes can appear noisy and irregular, posing a serious challenge to traditional, flow-based modeling techniques which assume locally smooth dynamics. To address this challenge, we propose MODE (Mixture Of Dynamical Experts), a graphical modeling framework whose neural gating mechanism decomposes complex dynamics into sparse, interpretable components, enabling both the unsupervised discovery of behavioral regimes and accurate long-term forecasting across regime transitions. Crucially, because agents in our framework can jump to different governing laws, MODE is especially tailored to the aforementioned noisy transitions. We evaluate our method on a battery of synthetic and real datasets from computational biology. First, we systematically benchmark MODE on an unsupervised classification task using synthetic dynamical snapshot data, including in noisy, few-sample settings. Next, we show how MODE succeeds on challenging forecasting tasks which simulate key cycling and branching processes in cell biology. Finally, we deploy our method on human, single-cell RNA sequencing data and show that it can not only distinguish proliferation from differentiation dynamics but also predict when cells will commit to their ultimate fate, a key outstanding challenge in computational biology.

Forecasting the long-term behavior of dynamical systems from sparse, noisy data is a major challenge in computational biology, applied physics and engineering (Bury et al., 2023; Moriel et al., 2024; Romeo et al., 2025). For example, predicting the developmental fates of progenitor cells from single cell RNA sequencing (scRNAseq) data is the subject of intense research in systems biology (Aivazidis et al., 2025). Two factors can make this endeavor especially difficult. First, data may consist of snapshots of unordered states and velocities at a single moment, $(x, \dot{x})$, instead of time series. This is often the case in transcriptomic, proteomic and other cellular data, which is acquired by destroying the tissue. Second, the dynamics generating the snapshots may consist of complex mixtures of overlapping behavioral regimes, so that forecasting rules learned for some samples do not generalize to others.

Consequently, a large body of research at the intersection of computational biology and data-driven dynamical systems has been devoted to the modeling of snapshot data. Standard approaches, like neural ordinary/partial differential equations (NODEs/NPDEs) (Lin et al., 2025), flow matching (Haviv et al., 2024; Wang et al., 2025) and dynamical optimal transport (Tong et al., 2020), attempt to fit a continuous-time flow in the form of an ODE or PDE which relates state to velocity. These are powerful, versatile approaches, but they can break down when overlapping dynamical regimes create ambiguous velocity signals. For example, when developing progenitor cells split into two different lineages (Fig. 1, left), cells of different fates (red vs green) can overlap in noisy transition zones where the dynamics are mixed (inset, with green cells having up-slanting velocities; red, down). Learning a single flow (e.g. Fig. 1, middle, NODE) tends to average the observed flows in this transition: the estimated developmental process then blurs between the two lineages (Fig. 1, middle, purple cells). The problem is all the more acute in cell cycle, where cycling and differentiating populations intermix (Mahdessian et al., 2021). This is a fundamental problem with all differential-

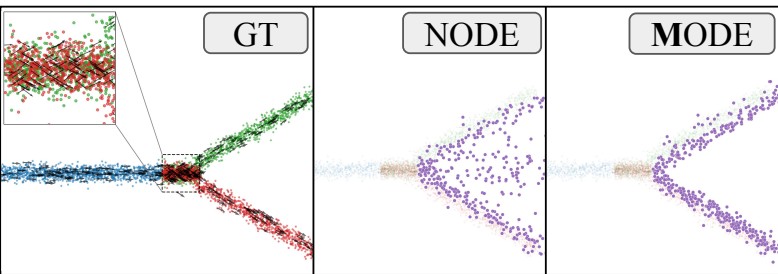

Figure 1: *Branching in in NODE vs MODE. (Left)* Ground truth progenitor cells in 2d advance along the blue trunk at a constant rate until they bifurcate into two fates, green and red, passing through an ambiguous switching region (inset; green cells with up-slanting velocities; red, with down-slanting). *(Middle)* Models that learn a single flow, like a NODE, can only reconcile switching zones by averaging. *(Right)* Instead, MODE learns a compositional flow with dedicated experts for trunk and branches, helping cells commit to their lineages.

equation modeling, which assumes trajectories change smoothly and cannot cross. With no means to disambiguate the underlying dynamical mixture, flow-based models struggle in these crucial cases.

Here, we tackle the case of branching, overlapping dynamics head-on using a neurally gated mixture of experts, MODE, (**M**ixture **O**f **D**ynamical **E**xperts) which learns compositional representations of complex flows. Unlike previous switching/hybrid models, MODE jointly learns dynamical regressors and mixture assignments on snapshot data, which allows circuit learning to influence regime discovery. As a result, it can tightly model the categorical branching choices which pervade single cell dynamics (Fig. 1, right, MODE). After training, MODE can be rolled out as a stochastic ODE in which evolving particles dynamically shift between different component dynamics. Importantly, we show that the number of experts can also be learned throughout training. The expert distribution can vary as a function of state, $x$, helping to model localized shifts in dynamics, or the experts can remain independent of $x$, in which case MODE can be used to classify heterogeneous mixtures of dynamical agents. Below, we demonstrate MODE's utility in both dynamical classification and forecasting tasks on synthetic and real data taken from across the biological sciences. Concretely, this paper's contributions are:

1. *Unsupervised classification of heterogeneous dynamical populations.* We first show how MODE can be used to classify mixtures of dynamical snapshots in an unsupervised manner. Data represent behaviors of intermingled dynamical agents with different governing equations. We show how MODE far outperforms standard unsupervised baselines and even competes with a supervised multi-layer-perceptron (MLP).

2. *Systematic noise and sample complexity benchmarking.* We run systematic tests on MODE's classification accuracy in increasingly difficult noise and sample complexity conditions, showing its favorable performance in these challenging and realistic cases.

3. *Forecasting in synthetic biological switches.* We then show how MODE outperforms existing methods on forecasting the long-term behavior of synthetic genetic switches representing fundamental biological processes, like the cell cycle and branching lineages.

4. *Predicting cycle exit in scRNAseq data.* Finally, we demonstrate how MODE can accurately detect, unsupervised, the cell cycle from scRNAseq data and that the mixture distribution can be used to quantitatively forecast the moment of cell differentiation and shed light on the transcriptional mechanisms driving this process.

## 1 RELATED WORK

**Data-driven dynamical systems.** Modeling dynamical systems from data has attracted considerable attention across applied mathematics, machine learning and computational biology (e.g. Schaffer & Kot 1985; Plum & Serra 2025; Khona & Fiete 2022; Costa et al. 2019; Bar et al. 2025). Traditional flow-based approaches, such as Neural Ordinary Differential Equations (NODEs) (Chen et al.,

2019) and other regressors like SINDy (Brunton et al., 2016), have proven powerful for learning continuous-time dynamics from data by parameterizing the derivative of hidden states through neural networks or symbolic regresison. However, they inherently assume a single, smooth flow, making them fundamentally unsuitable for systems exhibiting multiple coexisting dynamical regimes or abrupt transitions between behavioral modes.

**Flow learning in computational biology.** In computational biology, flow-based methods have found particular success in modeling cellular dynamics through RNA velocity estimation (Bergen et al., 2020) and optimal transport approaches (Tong et al., 2020). Meta Flow Matching (Atanackovic et al., 2024) represents a recent advancement, learning vector fields on the Wasserstein manifold to model population-level dynamics. Recent developments in structured latent velocity modeling (Farrell et al., 2023) have further extended these approaches by incorporating latent variable frameworks to capture complex single-cell transcriptomic dynamics, demonstrating improved performance in scenarios involving temporal gene expression patterns. While these approaches capture important aspects of biological systems, their modeling of branching lineages is completely deterministic. In essence, such modeling of the data assumes that the fate of cells is determined very early in the differentiation process, much before the actual branching into lineages. As such, these models do not capture the stochastic nature of biological processes.

**Switching dynamical systems.** The limitations of single-flow models have motivated extensive research into switching dynamical systems, which explicitly model transitions between different dynamical regimes. Classical approaches include piecewise affine models and hybrid systems (Jin et al., 2021), which segment trajectories and fit separate dynamics to each segment. More recent neural approaches (Ojeda et al., 2021; Seifner & Sanchez, 2023) have incorporated deep learning architectures to learn both the underlying dynamics and switching mechanisms simultaneously, often employing attention mechanisms or variational inference frameworks. Continuous-time switching systems (Köhs et al., 2021) extend these ideas by modeling regime transitions as Markov jump processes, providing principled probabilistic frameworks for handling temporal uncertainty. A related framework is found in Hybrid SINDy (Mangan et al., 2018), which pre-clusters data and fits cluster-specific dynamical regressors after the fact, enabling a switching dynamics on top of the basic SINDy framework.

While theoretically appealing for their ability to model multiple dynamics simultaneously they suffer from a reliance on trajectory (not snapshot) data or from their sequential (as opposed to joint) fitting of clusters and dynamics, which we will show below leads to suboptimal results for Hybrid SINDy.

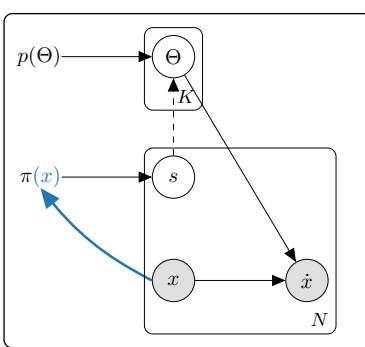

**Mixture of experts (MoE).** Mixture methods represent a fundamental approach throughout the data sciences. Early applications of MoE to dynamical systems identification (Lima et al., 2002; Weigend et al., 1995) demonstrated the ability to discover temporal regimes and avoid overfitting through soft partitioning of the input space. Unlike switching systems that make hard assignments, MoE models enable smooth transitions between experts through learned gating functions, making them particularly well-suited for noisy biological data where regime boundaries are inherently ambiguous. However, traditional MoEs typically lack interpretability in their expert components, as the experts are often represented by black-box neural networks. Furthermore, these methods very often rely on the availability of time series data, which is not the case of the snapshot data which are so prevalent in cell biology.

Figure 2: Plate diagram of MODE. Expert parameters $\Theta$ (with prior $p(\Theta)$) generate velocities $\dot{x}$ via $f_{\Theta_s}(x)$ under isotropic noise; the expert distribution optionally depends on $x$ (blue), in which case it gates states to specific experts, $s$.

Building on these foundations, our MODE framework unifies interpretable expert dynamics identification with the flexibility of mixture models. Unlike previous approaches that treat clustering and dynamics modeling as separate tasks, MODE jointly learns spatial organization and governing equations, enabling both accurate classification of heterogeneous populations and precise forecasting of regime transitions.

## 2 METHODS

Let $D = \{(x_i, \dot{x}_i)\}_{i=1}^N$, where $x_i \in \mathbb{R}^d$, denote snapshot data sampled from an underlying dynamical system. We assume the data are generated by $K$ latent dynamical laws, each with unknown governing equation $f_{\Theta_s}$ parameterized by $\Theta_s \in \mathbb{R}^m$. We place a Laplace prior $\Theta_s \sim \mathrm{Lap}(0, 1/\lambda)$ on the parameters, which promotes sparsity. Each data point is assigned to one of the $K$ experts, indexed by $s \in [K]$ which is categorically distributed according to the mixing distribution, $s \sim \pi$. The mixing distribution $\pi(x)$ may depend on the state $x$ (as in the cell cycle, where oscillatory regimes are localized in transcriptomic space) or may be independent of $x$ (as in the case of heterogeneous subpopulations with distinct intrinsic dynamics).

We model the probability of a state's velocity with isotropic Gaussian noise,

$$\dot{x} \mid x, s \sim \mathcal{N}\big(f_{\Theta_s}(x), \sigma_s^2 I_d\big).$$

The full generative model of the data (Fig. 2) is then

$$p(D \mid \boldsymbol{\Theta}, \sigma, \pi) = \prod_{i=1}^N \sum_{s=1}^K \pi_s(x_i)\, \mathcal{N}\big(\dot{x}_i \mid f_{\Theta_s}(x_i),\ \sigma_s^2 I_d\big), \tag{1}$$

where $\sigma = (\sigma_s)_{s=1}^K$, giving the conditional negative log likelihood on $\dot{x}$ as

$$-\log p(\dot{x} \mid x) = -\log \sum_{s=1}^K \pi_s(x)\, \exp\Big(-\tfrac{1}{2\sigma_s^2}\big\|\dot{x} - f_{\Theta_s}(x)\big\|_2^2 - \tfrac{d}{2}\log(2\pi\sigma_s^2)\Big). \tag{2}$$

Our goal is to estimate $\pi, \boldsymbol{\Theta}, \sigma$ by minimizing the maximum a posteriori (MAP) objective

$$\mathcal{L}(x, \dot{x}) = -\log \sum_{s=1}^K \pi_s(x)\, p(\dot{x} \mid x, s) + \lambda \sum_{s=1}^K \|\Theta_s\|_1. \tag{3}$$

There are several choices for how to model $f_{\Theta_s}$. In practice, we find that symbolic regression works well: $f_{\Theta_s}$ is computed as a linear combination of basis functions, $Z$, depending on $x$, so that

$$f_{\Theta_s}(x) = Z(x)\Theta_s,$$

where $\Theta_s$ serves as the weights. Throughout, we set $Z$ to be polynomials of order up to $c$, allowing for explicit control as to the complexity of the fitted approximation. In this form, MODE uses a SINDy-based regressor (Brunton et al., 2016) (See. App. Sec. C for details), though other estimators are possible (see Sec. 4). Here, we show results in which $K$ is known in advance, but we also provide an algorithm for discovering $K$ from data based on the Akaike (Akaike, 1974) or Bayesian Information Criteria (Schwarz, 1978) (AIC/ BIC; see Sec. Specifically, the algorithm iteratively evaluates whether adding or removing experts improves model fit while penalizing over-complexity. Starting from an initial estimate of K, the gating network's responsibilities are used to track expert usage, and redundant experts are pruned if their assignment drops below a threshold. This adaptive strategy ensures robust identification of the underlying number of dynamical regimes (see Sec. C.4), avoiding manual selection and reducing the risk of overfitting or underfitting.

If $\pi$ in Eq. 3 does not depend on $x$, then the MAP can be estimated efficiently using an expectation–maximization (EM) algorithm (see App. C.2). Otherwise, we set $\pi(x)$ to be a multilayer perceptron (MLP) in the manner of a gating network (Jacobs et al., 1991) and optimize Eq. 3 with stochastic gradient descent. However, when optimizing using gradient descent, the gating function has a tendency to converge to local minima corresponding to either a single expert or uniform probabilities everywhere. Thus, we found that performance was improved by regularizing Eq. 3 with the additional terms

$$H_{\text{gate}} = -\frac{1}{N} \sum_{i=1}^N \sum_{s=1}^K \pi_{i,s} \log \pi_{i,s} \quad ; \quad \mathrm{KL}_{\text{balance}} = \sum_{s=1}^K \bar{\pi}_s \left(\log \bar{\pi}_s - \log \tfrac{1}{K}\right) \tag{4}$$

where $H_{\text{gate}}$ encourages confident (low-entropy) gating decisions *per sample*, and $KL_{\text{balance}}$ prevents expert collapse by balancing average usage across experts, $\bar{\pi}_s$, across the dataset (see Sec.C). This

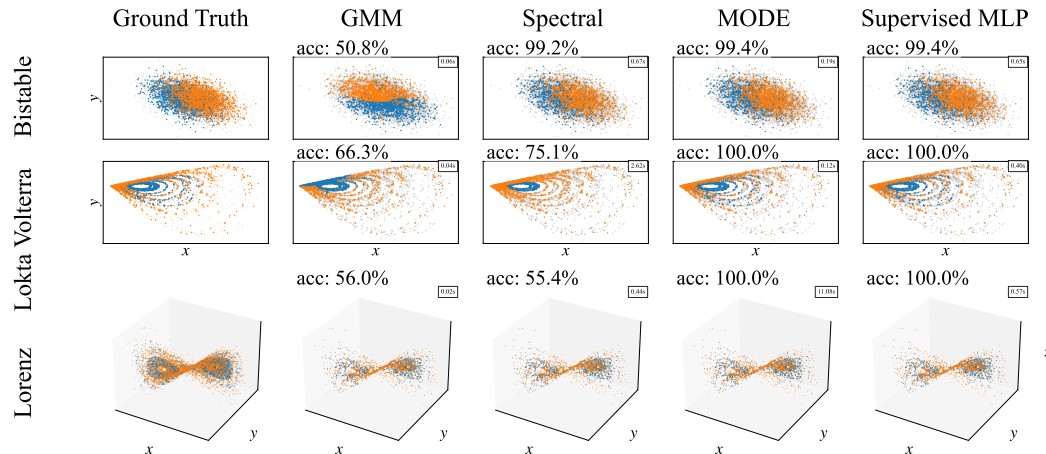

Figure 3: Elementary dynamics discovery on canonical systems. Each system challenges spatial clustering in distinct ways: **(Bistable)** spatially overlapping attractors with opposite rotational dynamics, **(Lotka-Volterra)** nonlinear predator-prey interactions creating complex phase structure, **(Lorenz)** chaotic flow with intricate spatial organization. Ground truth colors indicate true dynamical regimes. Clustering results colors indicate training data (gray) and estimated dynamical regimes (blue, orange). Accuracy of each method is displayed on the top-left of each subplots. Computation time is displayed at the upper right corner

combination of terms ensures that the gating network remains both selective and diverse, leveraging the strengths of individual experts while avoiding redundancy.

After training, we can simulate MODE forward in time according to the generative model, with a discrete-time Euler-Maruyama-like update:

$$s_{t+1} \sim \pi(\cdot \mid x_t) \tag{5}$$

$$x_{t+1} = x_t + \Delta t \, f_{\Theta_{s_{t+1}}}(x_t) + \sigma_B \sqrt{\Delta t} \, \xi_t, \tag{6}$$

where $\xi_t \sim \mathcal{N}(0, I_D)$ is standard $D$-dimensional Gaussian noise and $\sigma_B$ controls the stochasticity level. When $\pi$ is especially peaked, cells can "commit" to a new fate by sampling preferentially from that modal expert. Alternative rollout schemes are discussed in Sec. C.

## 3 RESULTS

### 3.1 DISENTANGLING HETEROGENEOUS DYNAMICAL POPULATIONS

Many complex systems in biology comprise heterogeneous mixtures of agents with different dynamical profiles, like tissues with multiple cell types or ecosystems with multiple species. To demonstrate that MODE's compositional dynamics provide essential advantages in disentangling these populations, we evaluate its performance on an unsupervised clustering task using mixture versions of three canonical dynamical systems where "geometry-only" methods systematically fail.

In particular, we generated data from a bistable attracting system (2d), the Lotka-Volterra predator-prey model (2d) and the Lorenz model of chaos (3d). In each case, snapshot samples, $(x, \dot{x})$, were randomly drawn from two distinct parameter settings, creating overlapping attracting blobs (bistable), concentric orbits (Lotka volterra) or intertwined paths (Lorenz) (Fig. 3, first column; see Sec.A for data details). Data consisted of snapshots $(x, \dot{x})$ with additive noise ($\sigma = 0.1$) split 80-20 into train and test. For each dataset we chose 2 sets of parameters and sampled following numerous initial conditions.

We fit a 2-MODE with $x$-independent mixture distributions to each of these data sets using an EM algorithm to optimize the expert allocation and dynamical parameters. We also fit a two-class gaussian mixture model (GMM) as well as a spectral clustering algorithm using standard, out-of-the-box parameters from scikit-learn. All models were trained with full snapshot data, $(x, \dot{x})$, including velocities. To ensure fair comparison, all methods were tuned via cross-validation, provided with

the correct number of clusters $K$, and evaluated using identical train-test splits (see Sec. C.2 for details). Additionally, we included a supervised MLP baseline using true cluster labels to establish an upper performance bound. After optimal cluster-to-class mapping, models were evaluated by their F1 score, recall, and precision.

| Method | Metric | Bistable | Lokta Volterra | Lorenz | Average |
|---|---|---|---|---|---|
| GMM | F1 | $0.527 \pm 0.000$ | $1.000 \pm 0.000$ | $0.523 \pm 0.000$ | $0.683 \pm 0.000$ |
| | Recall | $0.527 \pm 0.000$ | $1.000 \pm 0.000$ | $0.524 \pm 0.000$ | $0.684 \pm 0.000$ |
| | Precision | $0.527 \pm 0.000$ | $1.000 \pm 0.000$ | $0.524 \pm 0.000$ | $0.684 \pm 0.000$ |
| Spectral | F1 | $0.988 \pm 0.000$ | $0.765 \pm 0.032$ | $0.522 \pm 0.000$ | $0.758 \pm 0.011$ |
| | Recall | $0.988 \pm 0.000$ | $0.793 \pm 0.018$ | $0.523 \pm 0.000$ | $0.768 \pm 0.006$ |
| | Precision | $0.988 \pm 0.000$ | $0.864 \pm 0.003$ | $0.523 \pm 0.000$ | $0.792 \pm 0.001$ |
| MODE | F1 | $\mathbf{0.993} \pm 0.000$ | $\mathbf{1.000} \pm 0.000$ | $0.927 \pm 0.028$ | $0.974 \pm 0.009$ |
| | Recall | $\mathbf{0.993} \pm 0.000$ | $\mathbf{1.000} \pm 0.000$ | $0.927 \pm 0.028$ | $0.974 \pm 0.009$ |
| | Precision | $\mathbf{0.993} \pm 0.000$ | $\mathbf{1.000} \pm 0.000$ | $0.927 \pm 0.028$ | $0.974 \pm 0.009$ |
| MLP | F1 | $0.993 \pm 0.000$ | $\mathbf{1.000} \pm 0.000$ | $\mathbf{1.000} \pm 0.000$ | $\mathbf{0.997} \pm 0.000$ |
| | Recall | $0.993 \pm 0.000$ | $\mathbf{1.000} \pm 0.000$ | $\mathbf{1.000} \pm 0.000$ | $\mathbf{0.997} \pm 0.000$ |
| | Precision | $0.993 \pm 0.000$ | $\mathbf{1.000} \pm 0.000$ | $\mathbf{1.000} \pm 0.000$ | $\mathbf{0.997} \pm 0.000$ |

Table 1: Summary metrics: F1, Recall and Precision

We found that MODE achieved substantially higher F1 scores, recall, and precision than both unsupervised baselines and even competed with the supervised MLP (Table 1). Consistently, the GMM performed the worst, since it could only split ellipsoids in $(x, \dot{x})$ space, which fails dramatically for these thoroughly mixed systems. Spectral clustering could identify coherent bands from each population, but failed when the bands dissolved into salt-and-pepper noise. MODE, on the other hand, does not rely on phase space geometry, instead clustering systems according to the best division of data into two sparse, dynamical laws. This allows it to cluster populations which look like blobs, bands or true mixtures (Lorenz).

Thus, the improvement is most pronounced in cases where purely geometrical clustering fundamentally fails—precisely the scenarios most relevant to biological applications. For noise and sample size benchmarking, see Sec. B.2.

## 3.2 FORECASTING SYNTHETIC BIOLOGICAL SWITCHES

Although Sec. 3.1 demonstrates that MODE can be used to cluster dynamical agents with heterogeneous governing equations, it could be argued that this does not demonstrate MODE's value for modeling dynamics *per se*. After all, a key problem arising in computational biology and elsewhere is not just classifying snapshot data, but also forecasting its behavior into the future.

To that end, we sought to evaluate MODE's utility in forecasting the long-term behavior of dynamical systems with challenging switching behaviors. We examined this capability in two synthetic datasets modeling biological processes fundamental to all multi-cellular organisms: the cell cycle and cell lineage branching. Modeling these processes is made difficult by their inherent stochastic switching behaviors, whereby some cells randomly exit the cycle to continue their development or split off from a current lineage to join a new cell fate. We sought to examine how well MODE's mixture modeling could forecast this behavior compared to standard flow-learning baselines.

For cell cycle data, we used a classic three-variable model of mitotic oscillation from Goldbeter (Goldbeter, 1991) given by fluctuating levels of cyclin, maturation promoting factor (MPF) and protease (Fig. 4, top, first panel). We set the probability of cell cycle exit (differentiation, Fig. 4 top, first panel, orange branch) to be $p = 0.15$ in a small region on the cycle where cyclin was minimal. Cells which randomly committed to leaving the cycle in this region evolved according to a linear dynamics towards a stable node attractor (green X) placed about one cycle radius away. We sampled data uniformly in arc-length around both the cycle and exit path so that slow parts of the cycle would not be overrepresented. This resulted in 6320 positions and velocities separated into an 80-20 train-validation split.

For branching lineage data, we used a simple two-dimensional switching system (Fig. 4, bottom, first panel) similar to those used in dynamic optimal transport and flow matching studies of transcriptomic dynamics (Farrell et al., 2023; Zhang et al., 2024). A Gaussian blob of cells was advanced

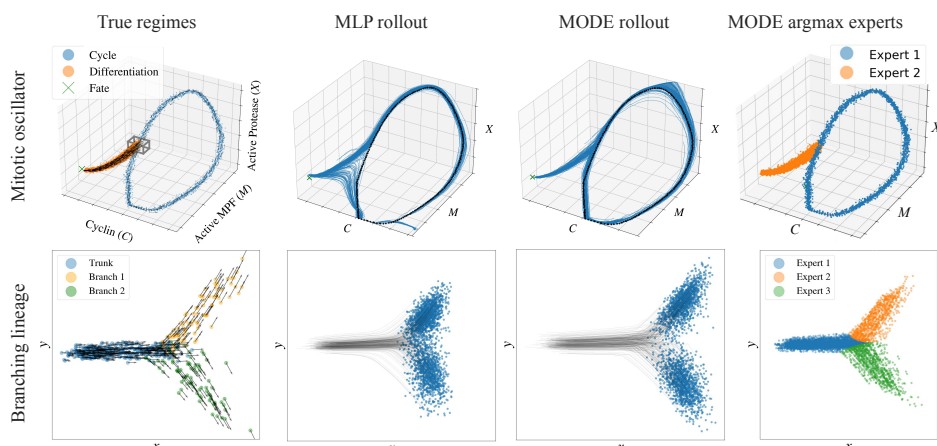

Figure 4: *MODE vs MLP baseline on synthetic biological switches. (Top row)* Cells evolving according to the Goldbeter oscillator (first panel, blue) could escape ($p = 0.15$ per time step) to a differentiated state (green X) when they pass through a region of low cyclin (gray box). The MLP (pictured, second panel) and SINDy try to use their single flow to explain the switching system, transiting near the differentiated branch's equilibrium before snaking back wildly towards the cycle. MODE (third panel) simply jettisons cells to the other expert with the appropriate probability since it has located the switching zone (argmax of $\pi(x)$, fourth panel). *(Bottom).* MODE's' performance advantage on simulated branching lineages (first panel) is similar: the MLP's (pictured, second panel) and SINDy's pushforward distribution blur the early split; MODE (third panel) better separates these branches since its experts locate the branches (fourth panel).

with a constant flow along a straight trunk (blue) for a fixed period. After traversing along the trunk, cells randomly branched (50-50) towards two different fates (orange, green) following two linear flows. The resulting 7500 samples were split 80-20 into train and validation.

**(a) Mitotic oscillator**

|  | $W_1$ | $W_2$ | $W_{1,C}$ | $W_{1,M}$ | $W_{1,X}$ |
|---|---|---|---|---|---|
| MLP | $0.3288 \pm 0.0012$ | $0.3759 \pm 0.0015$ | $0.2745 \pm 0.0009$ | $0.0896 \pm 0.0004$ | $0.1274 \pm 0.0006$ |
| SINDy | $0.5000 \pm 0.0020$ | $0.6000 \pm 0.0025$ | $0.4000 \pm 0.0015$ | $0.1500 \pm 0.0007$ | $0.2000 \pm 0.0010$ |
| Hybrid SINDy | $0.1097 \pm 0.0003$ | $0.1899 \pm 0.0005$ | $0.0312 \pm 0.0001$ | $0.0593 \pm 0.0002$ | $0.0706 \pm 0.0003$ |
| MODE-NN | $0.0913 \pm 0.0003$ | $0.1788 \pm 0.0005$ | $0.0294 \pm 0.0001$ | $0.0392 \pm 0.0002$ | $0.0695 \pm 0.0003$ |
| MODE-Symb | $\mathbf{0.0837 \pm 0.0005}$ | $\mathbf{0.1049 \pm 0.0010}$ | $\mathbf{0.0590 \pm 0.0003}$ | $\mathbf{0.0223 \pm 0.0002}$ | $\mathbf{0.0289 \pm 0.0002}$ |

**(b) Branching lineage**

|  | $W_1$ | $W_2$ | $W_{1,x}$ | $W_{1,y}$ |
|---|---|---|---|---|
| MLP | $0.6535 \pm 0.0018$ | $0.8371 \pm 0.0016$ | $\mathbf{0.1284 \pm 0.0006}$ | $0.6254 \pm 0.0015$ |
| SINDy | $0.9000 \pm 0.0025$ | $1.1000 \pm 0.0030$ | $0.2000 \pm 0.0010$ | $0.8000 \pm 0.0020$ |
| Hybrid SINDy | $0.6329 \pm 0.0017$ | $0.8200 \pm 0.0016$ | $0.1304 \pm 0.0006$ | $0.6053 \pm 0.0015$ |
| MODE-NN | $0.6124 \pm 0.0017$ | $0.8030 \pm 0.0016$ | $0.1324 \pm 0.0006$ | $0.5853 \pm 0.0015$ |
| MODE-Symb | $\mathbf{0.5713 \pm 0.0017}$ | $\mathbf{0.7689 \pm 0.0015}$ | $0.1363 \pm 0.0007$ | $\mathbf{0.5452 \pm 0.0014}$ |

Table 2: *Comparative performance of NODE vs baselines (see main text)* Wasserstein scores (lower is better) show distributional discrepancies between final states: $W_1$ (3D linear cost), $W_2$ (3D quadratic cost), and the marginal scores (per-coordinate 1D distances).

For the cycle data, we used a 2-expert MODE with up to cubic terms; for branching, a 3-expert MODE with linear terms. In these experiments, the mixture distribution for MODE depended on $x$ and a neural gating function was used, modeled as an MLP with one hidden layer. For baselines, we compared NODE with symbolic regressors (MODE-symb) to a (1) 4-layer MLP, (2) SINDy with cubic (linear) terms for the cycle (branching), (3) Hybrid SINDy (Mangan et al., 2018) using these same symbolic hyperparameters and MODE-NN in which the dynamical regressors were neural

ODEs. All models were trained with the same optimization and early-stopping criteria, and we confirmed both baselines could fit each dynamical component (i.e. Goldbeter cycle, cycle exit, branches) individually. The different models were compared to each other by pushing forward "progenitor cells" (from the cycle or trunk) under the estimated dynamics. The Wasserstein distance between the resulting positions of the cells and their groundtruth counterparts was calculated, using both the full joint and marginal distributions (full training details in Sec. C.3).

We found that MODE-Symb outperformed the baseline models on both tasks (Table 2). On the cycle data, the 2-expert MODE-Symb correctly identified the cell cycle exit (Fig. 4 top, fourth panel), and the probability assigned to the argmax expert in the exit region was close to the ground truth exit probability ($\pi_0(x) = 0.11 \pm 0.01$) (Fig. 13a). Single flow baselines (MLP, SINDy) tried to account for both the stable node and cycle (MLP, Fig. 4, top, second panel), which leads to high Wasserstein error. Hybrid SINDy struggled to find the true regimes during its pre-clustering phase and it never let cells exit the cycle. MODE-NN performed closest to MODE-Symb but seemed to suffer from the lack of sparsity constraint, which greatly aids in finding the true dynamical regimes (see App. Sec. C. Note that MODE-Symb still provides a very good fit of the Goldbeter oscillator, even though this system uses rational functions outside of our dictionary of basis functions.

On the branching data, MODE's stochastic rollout allowed it to commit (Fig. 4, bottom, third panel) more definitively to each branch, which decreased the error specifically in the $y$-dimensional marginal (i.e. the branching direction) (Tab. 2b). By contrast, baselines only captured the general quantitative behavior of the switch, spreading noncommittal cells in the branch interior. MODE-Symb experts correctly localized the branching structure (Fig. 4, bottom right) and discovered the underlying governing equations nearly exactly (see Sec. C.3).

## 3.3 FORECASTING CELL DEVELOPMENT FATE FROM SCRNASEQ DATA

The synthetic cellular processes examined in Sec. 3.2 were challenging to model for regular flow-learning approaches, but they were nevertheless highly idealized compared to their real, biological counterparts. In that spirit, we sought to understand if MODE could be used to forecast the developmental dynamics of the cell cycle from actual scRNAseq data. To that end, we analyzed single-cell RNA-sequencing data from the U2OS cell line from Mahdessian et al. in order to investigate the spatiotemporal dynamics of human cell cycle expression, chosen because this data contains a precise labeling for cells that are part of the cell-cycle versus those that are not. These labels give us a ground truth for two dynamical regimes that mimic our synthetic Goldbeter system of Sec. 3.2: the cycle regime, and the cycle exit regime. In Appendix Sec. B.6, we demonstrate similar results on an additional dataset of human fibroblasts.

Following a standard preprocessing protocol (Zheng et al., 2023), we used scVelo (Bergen et al., 2020) to estimate the local dynamics of each cell and gene, in the form of an RNA velocity (full details can be found in Appendix Sec. D.1). The gene expression data and the inferred velocities were projected into a 5-dimensional space using principal component analysis (PCA) for fitting (Fig. 5a).

We then fit this five-dimensional data with a 2-expert MODE having up to cubic terms, repeating this over ten random initializations to create an ensemble model (further training details in Sec. D.1). After training, we found that the ensemble averaged expert distribution (Fig. 5b) closely matched the true cell cycle scores (Fig. 5c), indicating that MODE's experts correctly divided gene expression space into a cycling program and a differentiation program. Additionally, Fig. 5d illustrates the stability of this segmentation, showing that all runs consistently converge to the correct number of regimes (cycling vs. differentiation).We also observed a characteristic entropic zone between these regimes (Fig. 5b, pink dots), acting as a stochastic escape route for differentiating cells to leave the cycle. We quantitatively confirmed MODE's fit of the true developmental dynamics by calculating the average ROC curve (Fig. 5d) from the ten training initializations, achieving an area-under-the-curve (AUC) score of 0.98, indicating a strong and stable correspondence to the true hidden regimes.

We also examined on this data whether the number of experts can be chosen adaptively during training. A metric relying on the Akaike Information Criterion (AIC) achieved by each expert was used in order to change the number of experts used (full details in Appendix Sec. C.4). Fig. 5e shows that our adaptive algorithm consistently converges at 2 experts, and that the AIC flattens out before the selection (Fig. 5f), indicating that it is a stable choice.

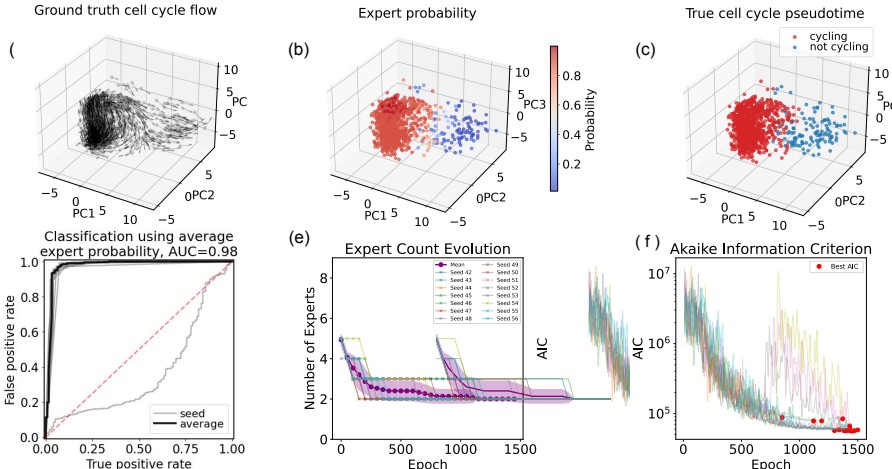

Figure 5: *Modeling cell-cycle and differentiation dynamics using MODE. (a)* Single cell RNA sequencing data from the U2OS bone cancer line was preprocessed for RNA velocity with scVelo (first three PCA dimensions shown). *(b)* We fit a 2-expert MODE to the first five PCA dimensions, revealing a sharp distinction between cycling and differentiating modes (red vs blue) averaged over ten random model initializations. *(c)* This closely matched ground truth scores computed from pseudotime, which allowed us to *(d)* accurately, and stably distinguish these two regimes. *(e)* Evolution of expert count over training epochs for 15 independent runs (colored lines) starting from $K = 5$ experts. The mean trajectory (purple line with markers) and confidence band (shaded region) show convergence to $K = 2$ experts, with most runs stabilizing by epoch 500. Individual runs demonstrate consistent adaptation despite different random initializations. *(f)* Akaike Information Criterion (AIC) trajectories for all runs. Each curve is truncated at its minimum AIC value (red dots), indicating the optimal model complexity. The log-scale y-axis shows rapid initial decrease followed by stabilization, confirming that the adaptive algorithm successfully identifies the optimal number of experts without manual tuning.

## 3.4 DISCOVERING CIRCUIT TRANSITIONS DURING CELL-CYCLE EXIT

To validate the biological relevance of MODE's learned dynamics, we analyzed expert-specific gene expression signatures and regulatory network rewiring during cell cycle exit. Differential expression analysis on experts (based on the analysis of the weighted Jacobian of the experts, see Appendix Sec. D.1) revealed that Expert 0 (cycling cells) upregulated canonical proliferation markers including TOP2A, CDK1, and AURKA (Malumbres & Barbacid, 2009), while Expert 1 (exiting cells) showed enrichment of exit-associated genes such as CAV1, THBS1, and CCN1(Coller et al., 2006) (Table 12 in Appendix Sec. D.1). Known G2/M markers (CCNB1, CDC20, CDK1) exhibited significantly higher expression in Expert 0, while exit markers (CAV1, CDKN1A) were enriched in Expert 1(Coller et al., 2006), confirming that MODE captures the known cycle exit phenotype.

Ordering cells by Expert 0 probability revealed coherent temporal expression patterns (Fig. 6A). Along this pseudotemporal trajectory, proliferation markers gradually decreased while exit markers increased, with the steepest changes occurring at the transition point defined by maximal expert entropy. This sequential pattern suggests coherent regulatory changes during exit that align with known molecular markers for cell cycle exit (Spencer et al., 2013).

To identify mechanistic drivers, we extracted gene regulatory networks (GRNs) from the experts by computing learned Jacobian matrices for data before and after cycle exit (see App. D.1 for methodology). GRN topology underwent substantial rewiring across transition phases (Fig. 6B). Before transition, the network featured prominent interactions among proliferative genes (CDK1–CCNB1–CDC20 module) (Malumbres & Barbacid, 2009). After transition, exit genes (CAV1, THBS1) gained regulatory centrality while proliferative interactions weakened. Notably, CCNB1 interactions shifted from promoting CDK1 activity to engaging exit-associated pathways, consistent with known checkpoint mechanisms(Pines, 1995).

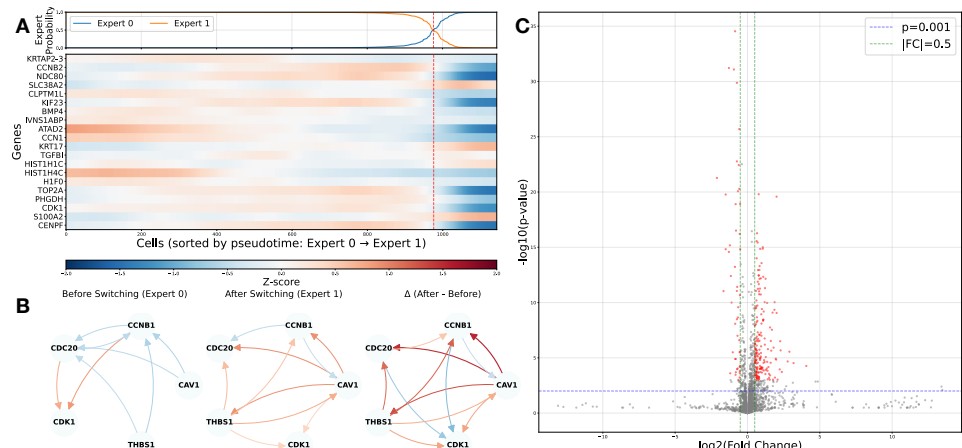

Figure 6: *Biological validation of cell cycle exit dynamics.* **A** Temporal heatmap of top differentially expressed genes along pseudotime (ordered by Expert 0 probability). Upper panel: expert probabilities; lower panel: gene expression z-scores. Red dashed line: transition point (maximum entropy). Proliferation markers decrease while exit markers increase. **B** Gene regulatory network topology before, during, and after transition. Circular graphs show key gene interactions (edge thickness: strength; color: sign). Bottom: differential connectivity. **C** Volcano plot of differential expression in high- vs low-entropy regions. Red: significant genes ($|FC| > 0.5, p < 0.001$). Dashed lines: thresholds.

Analysis of high-entropy cells associated with the moment of transition (top 20% by gating uncertainty) revealed 222 genes with significant expression changes ($|FC| > 0.5, p < 0.001$, Fig. 6C). Top transition-specific genes included CDC20B and PDK4, which differed from expert-defining markers, suggesting a distinct transcriptional state during fate resolution. These results provide evidence that MODE recovers both cell cycle regulators and mechanistic details of exit dynamics.

## 4    DISCUSSION AND FUTURE DIRECTIONS

Interacting agents in a real-world complex system can exhibit substantial dynamical variety, resulting from intrinsic biases, heterogeneous media, and differing noise levels. While single-flow methods can often explain the long-term behaviors of these heterogeneous systems on average, they can struggle in the case of strong branching, as we have shown. Building on earlier work in switching dynamical systems, MODE aims to discover these differing dynamical regimes from data. We showed the utility of this framework in unsupervised dynamical classification, forecasting in synthetic switches and the modeling of the cell cycle from scRNAseq data.

While these results are promising, MODE is currently limited by its lack of an explicit temporal component. Snapshot data is important and biologically relevant, but including dependence on time could help our framework learn more complex switching and branching schemes. For instance, the current stochastic rollout merely selects the most likely expert at each step. A more sophisticated approach could learn waiting times between switches, in the manner of a continuous-time Markov process. Extensions to higher dimensions, new regressors like neural ODEs, and hierarchical mixtures are also natural future directions. Furthermore, we are intrigued by the incorporation of augmented dynamical variables, like those in augmented neural ODEs (Dupont et al., 2019), which could be used in scRNAseq data to model effects like hidden proteometic influences during cell cycle dynamics.

Overall, we believe that the compositional representation of complex systems is a promising approach which captures the intuition that complicated phenomena should be broken down into simpler parts. MODE shows that this can be done in a straightforward way which can nevertheless outperform baselines on biologically-relevant problems.

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

## A  SYNTHETIC SYSTEMS

### A.1  DISCOVERING ELEMENTARY DYNAMICS IN SPATIALLY OVERLAPPING SYSTEMS

To systematically evaluate MODE's ability to discover elementary dynamics in complex systems, we constructed three canonical test cases that exhibit spatially overlapping regimes with distinct underlying dynamics (Figure 7). These synthetic systems serve as controlled benchmarks where ground truth cluster assignments are known, enabling rigorous quantitative assessment of clustering performance. Each system represents a different class of dynamical complexity commonly encountered in biological and physical applications, specifically designed to challenge spatial clustering methods while providing rich velocity information for dynamics-based approaches.

The **bistable system** (Figure 7, left panels) models two competing attractors that create spatially overlapping but dynamically distinct regimes. Data points are sampled from Gaussian distributions centered at $(-0.5, 0)$ and $(0.5, 0)$ with spread $\sigma = 1.0$, resulting in substantial spatial overlap between the two dynamical modes. The dynamics for each attractor follow:

$$\text{Mode 0:} \quad \dot{x} = -x + 2y, \quad \dot{y} = -0.5x - y - xy \quad (7)$$
$$\text{Mode 1:} \quad \dot{x} = -x - 2y, \quad \dot{y} = 0.5x - y + xy \quad (8)$$

where derivatives are computed relative to each attractor's center. As shown in the figure, while the two regimes occupy overlapping spatial regions, their velocity fields exhibit distinctly different flow patterns—Mode 0 displays clockwise rotation while Mode 1 shows counterclockwise dynamics. This configuration exemplifies systems where spatial clustering methods fail due to overlapping support, while velocity-based approaches can successfully distinguish the underlying dynamical regimes.

The **Lotka-Volterra system** (Figure 7, middle panels) implements predator-prey dynamics with two different parameter regimes, each generating distinct oscillatory behaviors. We simulate trajectories from 20 uniformly sampled initial conditions over the domain $[10, 50] \times [10, 50]$ and integrate each system for 100 time units. The two dynamical regimes are governed by:

$$\text{Mode 0:} \quad \dot{x} = 0.5x - 0.02xy, \quad \dot{y} = -0.5y + 0.01xy \tag{9}$$
$$\text{Mode 1:} \quad \dot{x} = 0.5x - 0.04xy, \quad \dot{y} = -0.6y + 0.01xy \tag{10}$$

The different parameter values create distinct cycle shapes and periods, as visualized by the concentric orbits in the phase portraits. The resulting trajectories exhibit overlapping spatial support where position alone cannot distinguish between regimes, yet the velocity patterns remain characteristic of each underlying dynamics, with Mode 0 generating tighter elliptical cycles and Mode 1 producing broader, more circular orbits.

The **Lorenz system** (Figure 7, right panels) represents chaotic dynamics with two distinct parameter sets, each producing different attractor geometries. We generate trajectories from 20 random initial conditions uniformly distributed over $[-15, 15] \times [-15, 15] \times [0, 40]$ and integrate for 10 time units, discarding the first 1000 time steps to eliminate transients. The two chaotic regimes follow:

$$\text{Mode 0:} \quad \dot{x} = 12(y - x), \quad \dot{y} = x(28 - z) - y, \quad \dot{z} = xy - 4z \tag{11}$$

$$\text{Mode 1:} \quad \dot{x} = 10(y - x), \quad \dot{y} = x(35.65 - z) - y, \quad \dot{z} = xy - \frac{8z}{3} \tag{12}$$

These parameter choices generate attractors with different wing shapes and temporal dynamics. As illustrated in the three-dimensional projections, trajectory segments from different regimes occupy similar spatial regions but exhibit distinct local flow patterns, creating a challenging clustering scenario where velocity information provides the crucial discriminating signal.

For all systems, we add Gaussian noise with standard deviation $\sigma = 0.1$ to both position and velocity measurements to simulate realistic experimental conditions. Each dataset contains $n = 10,000$ data points equally distributed between the two dynamical regimes. The resulting datasets exhibit the key challenge that motivates MODE: spatial clustering methods fail due to overlapping support, while velocity information provides the crucial distinguishing signal needed for accurate regime identification. This experimental design directly tests MODE's central hypothesis that incorporating local dynamics can resolve ambiguities that spatial methods cannot handle.

## A.2 FORECASTING SYNTHETIC BIOLOGICAL SWITCHES

To evaluate MODE's forecasting capabilities on biologically relevant switching dynamics, we constructed two synthetic datasets that capture fundamental processes in cellular biology: cell cycle dynamics with stochastic exit events and lineage branching processes. These systems represent challenging forecasting scenarios where single-flow models fundamentally fail due to the presence of discrete regime transitions and branching trajectories that violate the assumptions of continuous dynamical systems.

**Cell cycle dataset with differentiation exit.** The cell cycle dataset implements the classic three-variable Goldbeter oscillator model (Goldbeter, 1991), which describes mitotic oscillations through the temporal dynamics of cyclin concentration, maturation promoting factor (MPF), and protease activity. The Goldbeter model captures essential features of cell cycle regulation through a system of nonlinear ordinary differential equations that generate stable limit cycle behavior representing normal cell division cycles. These are given by:

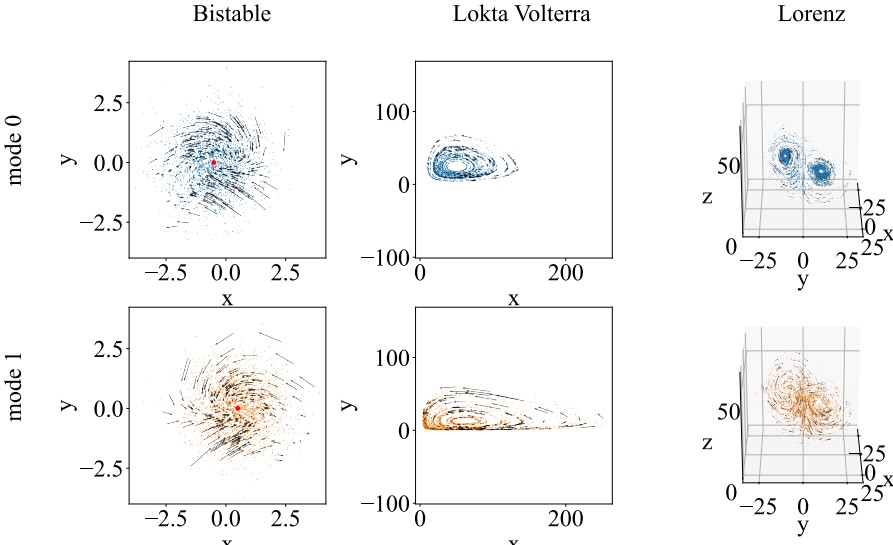

Figure 7: **Synthetic datasets for elementary dynamics discovery benchmarks.** Phase portraits showing spatially overlapping dynamical regimes for three canonical test systems. Each column represents a different system (bistable, Lotka-Volterra, Lorenz), with Mode 0 (top row, blue) and Mode 1 (bottom row, orange) displaying distinct velocity fields despite spatial overlap. Arrows indicate local flow directions on a subsample of data points. **Left:** Bistable system with competing attractors showing clockwise vs. counterclockwise rotation patterns. **Middle:** Lotka-Volterra predator-prey dynamics with different oscillatory regimes producing distinct cycle shapes. **Right:** Lorenz chaotic system with different parameter sets generating contrasting attractor geometries in 3D projections. These datasets demonstrate the fundamental challenge MODE addresses: spatial clustering fails due to overlapping support, while velocity information provides the discriminating signal for accurate regime identification.

$$\frac{dC}{dt} = v_i - k_d C - \frac{v_d\, X\, C}{K_d + C} \tag{13}$$

$$\frac{dM}{dt} = \frac{V_1(1 - M)}{K_1 + (1 - M)} - \frac{V_2 M}{K_2 + M}, \quad V_1 = \frac{V_{M1}\, C}{K_c + C} \tag{14}$$

$$\frac{dX}{dt} = \frac{V_3(1 - X)}{K_3 + (1 - X)} - \frac{V_4 X}{K_4 + X}, \quad V_3 = V_{M3}\, M \tag{15}$$

Parameters were set to the standard values found in Table 3.

We augmented the standard Goldbeter oscillator with a stochastic differentiation exit mechanism to simulate the biological phenomenon where cycling cells can irreversibly commit to differentiation. Specifically, we defined a spatial exit zone in the region of minimal cyclin concentration (the G1/S checkpoint region) where cells have a probability $p = 0.15$ per time step of exiting the cell cycle. Cells that stochastically transition in this region evolve according to linear dynamics toward a stable node attractor representing the differentiated state, positioned approximately one cycle radius away from the oscillator trajectory. Specifically, cells evolved according to

$$\frac{d\mathbf{y}}{dt} = -K\left(\mathbf{y} - \mathbf{x}^*\right), \tag{16}$$

where

$$\mathbf{x}^* = \begin{bmatrix} -0.3461 \\ 0.1481 \\ 0.1468 \end{bmatrix}, \qquad K = \begin{bmatrix} 0.6000 & 0 & 0 \\ 0 & 0.8000 & 0 \\ 0 & 0 & 0.9000 \end{bmatrix}.$$

| Symbol | Code parameter | Value |
|--------|----------------|-------|
| $v_i$ | vi | 0.0335 |
| $k_d$ | kd | 0.0000 |
| $v_d$ | vd | 0.2500 |
| $K_d$ | Kd | 0.0200 |
| $V_{M1}$ | VM1 | 3.2523 |
| $K_c$ | Kc | 0.5000 |
| $K_1$ | K1 | 0.0050 |
| $V_2$ | V2 | 0.8158 |
| $K_2$ | K2 | 0.0050 |
| $V_{M3}$ | VM3 | 1.7020 |
| $K_3$ | K3 | 0.0050 |
| $V_4$ | V4 | 1.1580 |
| $K_4$ | K4 | 0.0050 |

Table 3: Employed parameters used for Goldbeter's mitotic oscillator.

The resulting dynamics combine three distinct behavioral regimes: (1) stable oscillatory behavior along the limit cycle, (2) stochastic transition events in the exit zone, and (3) deterministic approach to the differentiated attractor. This configuration creates a challenging forecasting scenario where successful prediction requires accurately modeling both the continuous cycle dynamics and the discrete exit decisions that fundamentally alter cell fate trajectories.

Data generation involved uniform sampling in arc-length along both the cycle trajectory and the differentiation path to ensure balanced representation across dynamical regimes and prevent over-sampling of slow trajectory segments. This sampling strategy produced $n = 6,320$ position-velocity pairs $(x_i, \dot{x}_i)$ with additive Gaussian noise ($\sigma = 0.1$), split into 80% training and 20% validation sets. The ground truth exit probability in the transition zone provides quantitative validation of MODE's learned gating function.

**Lineage branching dataset.** The branching dataset implements a simplified model of cellular lineage specification commonly observed in developmental biology, where a homogeneous progenitor population splits into distinct cell fates through binary decision processes. This system models the essential features of lineage commitment: initial homogeneous dynamics followed by irreversible branching toward distinct terminal states.

Specifically, $n = 600$ initial cells were drawn from a Gaussian distribution with parameters

$$\boldsymbol{\mu}_0 = \begin{bmatrix} 0 \\ 0 \end{bmatrix}, \qquad \Sigma_0 = 0.08\, I_2.$$

These were evolved according to the system

$$\dot{\mathbf{x}} = A_T \mathbf{x} + \mathbf{c}, \qquad A_T = \begin{bmatrix} 0.15 & -0.05 \\ 0.05 & 0.10 \end{bmatrix}, \quad \mathbf{c} = \begin{bmatrix} 0.6 \\ 0 \end{bmatrix},$$

for 45 steps, at which point 50% of cells were divided between the two lineages.

These new populations evolved according to

$$\dot{\mathbf{x}} = \begin{bmatrix} 0 & 0 \\ \pm s & 0 \end{bmatrix} \mathbf{x} + \mathbf{c}, \qquad s = 0.6,$$

for 30 steps. Throughout, the step size was given by $\Delta t = 0.08$. By sampling all trajectories every step, this process accumulated a total of $600 \times (45 + 30) = 45{,}000$ samples, divided into an 80–20 train-validation split.

**Experimental design and evaluation metrics.** Both datasets challenge forecasting methods through distinct mechanisms: the cell cycle system tests handling of stochastic regime transitions within continuous dynamics, while the branching system evaluates prediction of discrete fate decisions and trajectory divergence. These complementary challenges provide comprehensive assessment of MODE's forecasting capabilities across the spectrum of switching behaviors encountered in biological systems.

Evaluation employed pushforward analysis, where "progenitor cells" sampled from initial conditions (cycle initiation points or trunk origins) were evolved under learned dynamics and compared to ground truth trajectories using Wasserstein distance metrics. This approach directly measures forecasting accuracy for the biological phenomena of interest: predicting cell cycle exit timing and lineage commitment outcomes. Both full joint distributions and marginal distributions were evaluated to assess different aspects of forecasting performance, with particular attention to accuracy in the branching dimension for lineage specification tasks.

# B    Additional Experimental Results

## B.1    Governing equation recovery in multi-modal dynamical systems

Beyond clustering trajectories into distinct dynamical regimes, MODE simultaneously recovers the governing equations for each discovered mode. This dual capability enables mechanistic interpretation of multi-modal systems by providing sparse polynomial representations of the underlying dynamics. We evaluate MODE's equation discovery performance on three canonical test systems, comparing recovered coefficients against ground truth dynamics.

For each system, we apply MODE with polynomial feature libraries of degree 2, enabling simultaneous trajectory clustering and sparse coefficient identification. The recovered equations provide direct access to the mathematical structure distinguishing different dynamical regimes, offering interpretability that pure clustering methods cannot provide.

**Bistable system analysis.**    The bistable system exhibits spatially overlapping attractors with nonlinear coupling terms that create distinct basins of attraction. Table 4 reveals MODE's capacity for precise coefficient recovery across both modes. The method successfully identifies all structurally important coefficients while maintaining sparsity by correctly setting zero-valued terms to negligible values.

MODE accurately recovers the critical dynamical structure: Mode 0 follows $\dot{x} = -0.5 - x + 2y$ and $\dot{y} = -0.25 - 0.5x - 1.5y - xy$, while Mode 1 exhibits $\dot{x} = 0.5 - x - 2y$ and $\dot{y} = -0.25 + 0.5x - 1.5y + xy$. The sign reversal in both linear and nonlinear terms creates the bistable character, and MODE's precise identification of these differences (errors $\mathcal{O}(10^{-5})$) demonstrates its sensitivity to dynamical rather than geometric features.

**Lotka-Volterra system analysis.**    The predator-prey dynamics present temporal rather than spatial mode separation, challenging traditional clustering approaches. Table 5 demonstrates MODE's ability to resolve distinct ecological parameter regimes within the same phase space region. The method correctly recovers the prey growth rates ($0.5$ in both modes) and distinguishes the predation efficiencies: $0.02$ for Mode 0 versus $0.04$ for Mode 1.

Most critically, MODE identifies the different prey mortality parameters that characterize each regime: $-0.5$ for Mode 0 and $-0.6$ for Mode 1. These parameter differences, while seemingly modest, create qualitatively distinct cyclical behaviors that MODE successfully captures through its velocity-informed clustering approach. Coefficient estimation errors remain at the level of $\mathcal{O}(10^{-7})$ for the principal terms.

**Lorenz system analysis.**    The chaotic Lorenz attractor represents the most challenging case for equation discovery, with three-dimensional dynamics and sensitive dependence on initial conditions. Table 6 reveals both the capabilities and limitations of MODE's approach in chaotic regimes. While the method successfully recovers the fundamental structure of the Lorenz equations, estimation accuracy varies significantly across coefficient types.

MODE accurately identifies the critical nonlinear coupling terms: the $xy$ coefficients in the $\dot{z}$ equations (recovered as $1.00 \pm 10^{-4}$ and $1.01 \pm 10^{-3}$) and maintains the essential $xz$ coupling structure with coefficients of $-0.999 \pm 10^{-6}$ and $-1.00 \pm 10^{-5}$. However, the method exhibits larger errors in constant and linear terms, particularly evident in the $\dot{y}$ equations where estimated constant terms ($1.29 \pm 10^{1}$ and $-1.59 \pm 10^{1}$) deviate substantially from the true values of 0.

Table 4: Equation recovery for the bistable system. Each cell shows the true coefficient, estimated mean $\pm$ variance over 10 runs, and absolute error. MODE achieves near-perfect recovery with errors at the level of $\mathcal{O}(10^{-6})$.

|  | 1 | $\hat{1}$ | $x$ | $\hat{x}$ | $y$ | $\hat{y}$ |
|---|---|---|---|---|---|---|
| mode 0 dx/dt | $-0.5$ | $-0.493 \pm 10^{-5}$ | $-1$ | $-0.988 \pm 10^{-5}$ | 2 | $1.98 \pm 10^{-5}$ |
| mode 0 dy/dt | $-0.25$ | $-0.248 \pm 10^{-5}$ | $-0.5$ | $-0.497 \pm 10^{-5}$ | $-1.5$ | $-1.49 \pm 10^{-5}$ |
| mode 1 dx/dt | 0.5 | $0.497 \pm 10^{-5}$ | $-1$ | $-0.992 \pm 10^{-6}$ | $-2$ | $-1.98 \pm 10^{-5}$ |
| mode 1 dy/dt | $-0.25$ | $-0.244 \pm 10^{-5}$ | 0.5 | $0.491 \pm 10^{-5}$ | $-1.5$ | $-1.48 \pm 10^{-4}$ |

|  | $x^2$ | $\hat{x^2}$ | $xy$ | $\hat{xy}$ | $y^2$ | $\hat{y^2}$ |
|---|---|---|---|---|---|---|
| mode 0 dx/dt | 0 | $-0.000152 \pm 10^{-5}$ | 0 | $0.00262 \pm 10^{-5}$ | 0 | $-0.00307 \pm 10^{-6}$ |
| mode 0 dy/dt | 0 | $0.000716 \pm 10^{-5}$ | $-1$ | $-0.987 \pm 10^{-5}$ | 0 | $-0.00317 \pm 10^{-6}$ |
| mode 1 dx/dt | 0 | $0.00474 \pm 10^{-6}$ | 0 | $-0.000486 \pm 10^{-5}$ | 0 | $-0.000675 \pm 10^{-5}$ |
| mode 1 dy/dt | 0 | $-0.00102 \pm 10^{-5}$ | 1 | $0.98 \pm 10^{-5}$ | 0 | $-0.000296 \pm 10^{-5}$ |

Table 5: Equation recovery for the Lotka-Volterra predator-prey system. MODE correctly identifies distinct predation rates and mortality parameters that characterize each ecological regime.

|  | 1 | $\hat{1}$ | $x$ | $\hat{x}$ | $y$ | $\hat{y}$ |
|---|---|---|---|---|---|---|
| mode 0 dx/dt/dt | 0 | $-0.0281 \pm 10^{-4}$ | 0.5 | $0.501 \pm 10^{-7}$ | 0 | $0.000761 \pm 10^{-6}$ |
| mode 0 dy/dt/dt | 0 | $-0.006 \pm 10^{-4}$ | 0 | $-2.17e-05 \pm 10^{-7}$ | $-0.5$ | $-0.5 \pm 10^{-7}$ |
| mode 1 dx/dt/dt | 0 | $0.00661 \pm 10^{-4}$ | 0.5 | $0.499 \pm 10^{-7}$ | 0 | $0.000474 \pm 10^{-7}$ |
| mode 1 dy/dt/dt | 0 | $0.00118 \pm 10^{-5}$ | 0 | $-4.09e-05 \pm 10^{-8}$ | $-0.6$ | $-0.6 \pm 10^{-7}$ |

|  | $x^2$ | $\hat{x^2}$ | $xy$ | $\hat{xy}$ | $y^2$ | $\hat{y^2}$ |
|---|---|---|---|---|---|---|
| mode 0 dx/dt/dt | 0 | $-3.75e-07 \pm 10^{-11}$ | $-0.02$ | $-0.02 \pm 10^{-11}$ | 0 | $2.34e-06 \pm 10^{-10}$ |
| mode 0 dy/dt/dt | 0 | $-9.74e-08 \pm 10^{-11}$ | 0.01 | $0.01 \pm 10^{-11}$ | 0 | $-2.98e-06 \pm 10^{-10}$ |
| mode 1 dx/dt/dt | 0 | $3.02e-06 \pm 10^{-11}$ | $-0.04$ | $-0.04 \pm 10^{-10}$ | 0 | $-3.12e-05 \pm 10^{-9}$ |
| mode 1 dy/dt/dt | 0 | $6.79e-07 \pm 10^{-12}$ | 0.01 | $0.00999 \pm 10^{-11}$ | 0 | $4.6e-06 \pm 10^{-10}$ |

These larger estimation errors in the Lorenz system reflect the inherent challenges of applying sparse regression to chaotic data. The sensitive dependence on initial conditions and the presence of multiple time scales create difficulties in cleanly separating trajectories between modes, leading to coefficient bias when the clustering assignment itself contains uncertainty. Despite these limitations, MODE successfully captures the essential nonlinear structure that governs the chaotic dynamics.

**Statistical consistency and limitations.** All results represent averages over 10 independent runs with different random initializations, providing estimates of method reliability. For the bistable and Lotka-Volterra systems, coefficient variances remain extremely low ($\mathcal{O}(10^{-7})$ or smaller), indicating consistent convergence across runs. The Lorenz system shows higher variance, particularly for constant terms, reflecting the added complexity of chaotic trajectory separation.

The equation recovery demonstrates MODE's interpretability advantage: unlike black-box clustering methods, MODE provides explicit mathematical models revealing the mechanisms underlying each dynamical regime. This capability transforms trajectory clustering from pattern recognition into scientific discovery, enabling researchers to understand not just which trajectories belong together, but why they exhibit similar dynamics.

### B.2 NOISE AND TRAINING SIZE ROBUSTNESS

To assess the practical applicability of elementary dynamics decomposition, we conduct systematic robustness studies examining how clustering performance degrades under realistic experimental conditions. Real-world dynamical systems inevitably contain measurement noise and are often constrained by limited sample sizes, factors that can severely impact clustering methods that rely on precise velocity estimates.

Table 6: Equation recovery for the Lorenz chaotic system. Despite the system's complexity and three-dimensional dynamics, MODE accurately recovers the key parameters distinguishing the two chaotic attractors.

| | $1$ | $\hat{1}$ | $x$ | $\hat{x}$ | $y$ | $\hat{y}$ |
|---|---|---|---|---|---|---|
| mode 0 dx/dt/dt | 0 | $0.0993 \pm 10^{-2}$ | $-12$ | $-12 \pm 10^{-3}$ | 12 | $11.9 \pm 10^{-2}$ |
| mode 0 dy/dt/dt | 0 | $1.29 \pm 10^{1}$ | 28 | $28.5 \pm 10^{0}$ | $-1$ | $-1.1 \pm 10^{-2}$ |
| mode 0 dz/dt/dt | 0 | $-0.0777 \pm 10^{-3}$ | 0 | $-0.653 \pm 10^{0}$ | 0 | $0.266 \pm 10^{-1}$ |
| mode 1 dx/dt/dt | 0 | $0.0523 \pm 10^{-2}$ | $-10$ | $-10.2 \pm 10^{-1}$ | 10 | $10.1 \pm 10^{-1}$ |
| mode 1 dy/dt/dt | 0 | $-1.59 \pm 10^{1}$ | 35.6 | $35.4 \pm 10^{-1}$ | $-1$ | $-1.13 \pm 10^{-1}$ |
| mode 1 dz/dt/dt | 0 | $-0.0963 \pm 10^{-2}$ | 0 | $0.553 \pm 10^{0}$ | 0 | $-0.212 \pm 10^{-1}$ |

| | $z$ | $\hat{z}$ | $x^2$ | $\hat{x^2}$ | $xy$ | $\hat{xy}$ |
|---|---|---|---|---|---|---|
| mode 0 dx/dt/dt | 0 | $0.0386 \pm 10^{-2}$ | 0 | $0.00456 \pm 10^{-4}$ | 0 | $0.00152 \pm 10^{-5}$ |
| mode 0 dy/dt/dt | 0 | $-0.299 \pm 10^{-1}$ | 0 | $-0.0516 \pm 10^{-2}$ | 0 | $0.0279 \pm 10^{-3}$ |
| mode 0 dz/dt/dt | $-4$ | $-3.92 \pm 10^{-2}$ | 0 | $-0.00421 \pm 10^{-4}$ | 1 | $1 \pm 10^{-4}$ |
| mode 1 dx/dt/dt | 0 | $-0.0406 \pm 10^{-2}$ | 0 | $-0.00311 \pm 10^{-4}$ | 0 | $-0.002 \pm 10^{-5}$ |
| mode 1 dy/dt/dt | 0 | $0.265 \pm 10^{-1}$ | 0 | $0.0532 \pm 10^{-2}$ | 0 | $-0.0362 \pm 10^{-2}$ |
| mode 1 dz/dt/dt | $-2.67$ | $-2.76 \pm 10^{-2}$ | 0 | $-0.0173 \pm 10^{-3}$ | 1 | $1.01 \pm 10^{-3}$ |

| | $xz$ | $\hat{xz}$ | $y^2$ | $\hat{y^2}$ | $yz$ | $\hat{yz}$ |
|---|---|---|---|---|---|---|
| mode 0 dx/dt/dt | 0 | $0.000759 \pm 10^{-6}$ | 0 | $-0.000665 \pm 10^{-6}$ | 0 | $-0.000515 \pm 10^{-6}$ |
| mode 0 dy/dt/dt | $-1$ | $-0.999 \pm 10^{-6}$ | 0 | $-0.0054 \pm 10^{-4}$ | 0 | $0.00202 \pm 10^{-5}$ |
| mode 0 dz/dt/dt | 0 | $0.0094 \pm 10^{-4}$ | 0 | $-0.000259 \pm 10^{-6}$ | 0 | $-0.00324 \pm 10^{-5}$ |
| mode 1 dx/dt/dt | 0 | $0.000664 \pm 10^{-6}$ | 0 | $0.000369 \pm 10^{-7}$ | 0 | $-0.000614 \pm 10^{-6}$ |
| mode 1 dy/dt/dt | $-1$ | $-1 \pm 10^{-5}$ | 0 | $0.011 \pm 10^{-3}$ | 0 | $0.00326 \pm 10^{-5}$ |
| mode 1 dz/dt/dt | 0 | $-0.00809 \pm 10^{-4}$ | 0 | $-0.00252 \pm 10^{-5}$ | 0 | $0.00249 \pm 10^{-5}$ |

| | $z^2$ | $\hat{z^2}$ |
|---|---|---|
| mode 0 dx/dt/dt | 0 | $-0.00187 \pm 10^{-5}$ |
| mode 0 dy/dt/dt | 0 | $0.0076 \pm 10^{-4}$ |
| mode 0 dz/dt/dt | 0 | $0.000345 \pm 10^{-6}$ |
| mode 1 dx/dt/dt | 0 | $0.00174 \pm 10^{-5}$ |
| mode 1 dy/dt/dt | 0 | $-0.00652 \pm 10^{-4}$ |
| mode 1 dz/dt/dt | 0 | $0.00218 \pm 10^{-5}$ |

We evaluated robustness across three canonical test systems (bistable, Lotka-Volterra, Lorenz) by systematically varying two critical experimental parameters: Gaussian noise levels ($\sigma \in \{0.0, 10^{-5}, 10^{-2}, 10^{-1}, 2 \times 10^{-1}\}$) and training dataset sizes ($N \in \{100, 800, 8000\}$). Noise is added to both position $x$ and velocity $\dot{x}$ measurements after normalization ($VAR[x] = VAR[\dot{x}] = 1$) to simulate realistic experimental conditions. For each parameter combination, we evaluate clustering performance using ARI and NMI metrics on held-out test data (20% of each dataset), averaging results over multiple random seeds to ensure statistical reliability.

Figure 8 reveals that MODE demonstrates competitive performance compared to the MLP baseline while substantially outperforming spatial-only methods across most test conditions. MODE shows robust performance on the bistable and Lotka-Volterra systems, maintaining high clustering accuracy (ARI, NMI > 0.8) even under moderate noise levels where spatial clustering methods fail completely. This advantage stems from MODE's ability to leverage velocity information—even noisy velocity estimates provide directional information that purely spatial clustering cannot access.

The noise robustness patterns reveal system-specific insights: the bistable system shows the most dramatic performance advantages for MODE over spatial methods, with MODE maintaining high accuracy while GMM and Spectral clustering fail completely across most noise levels. This occurs because the bistable system's spatially overlapping attractors become indistinguishable without velocity information as noise increases. The Lorenz system presents the most challenging case, where all methods show performance degradation, though MODE and MLP maintain more robust performance than spatial-only approaches.

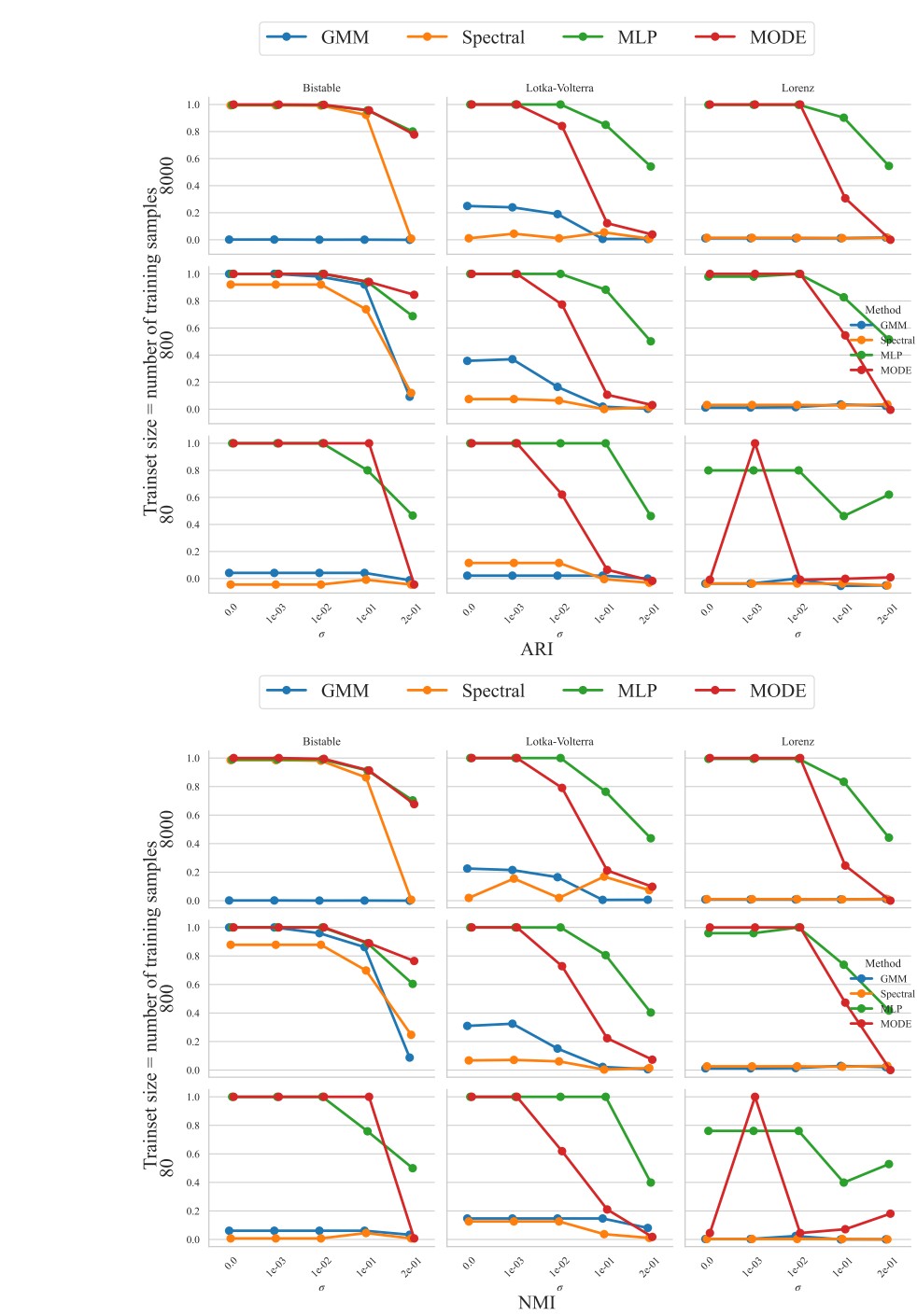

Figure 8: Clustering performance (ARI, NMI) as a function of noise level ($\sigma$) and training set size ($N$) across different dynamical systems. Each subplot corresponds to a specific dataset and metric. MODE demonstrates superior noise tolerance and sample efficiency compared to spatial-only methods, maintaining high performance even under challenging experimental conditions where baseline methods fail.

The training size analysis reveals differential sample efficiency across systems. On the bistable and Lotka-Volterra systems, MODE maintains reasonable clustering performance even with limited samples ($N = 100$), while spatial methods show severe degradation in small-sample regimes. However,

performance variability increases substantially at the smallest sample size ($N = 100$), indicating that while MODE can work with limited data, larger sample sizes ($N \geq 800$) provide more reliable results.

The Lotka-Volterra system demonstrates MODE's effectiveness for temporally-separated dynamics, where spatial clustering methods consistently fail regardless of sample size. The underlying advantage is that velocity information provides crucial directional cues about cluster membership that spatial position alone cannot capture—each $(x_i, \dot{x}_i)$ pair encodes both current state and dynamical tendency, enabling more robust clustering even when phase space regions overlap.

These robustness results provide practical guidance for experimental applications. MODE's noise tolerance suggests applicability to real biological data with moderate measurement uncertainties, though performance degrades significantly under high noise conditions ($\sigma > 0.1$). The sample size analysis indicates that while MODE can function with limited data, reliable performance requires adequate sample sizes ($N \geq 800$ recommended) to ensure consistent clustering results.

The comparison with the MLP baseline reveals an important trade-off: while MLP often matches or slightly exceeds MODE's clustering performance, MODE provides the critical advantage of interpretable sparse dynamics. MODE achieves competitive robustness while simultaneously offering mechanistic insight through its recovered governing equations, making it particularly suitable for scientific applications where understanding system structure is as important as clustering accuracy.

### B.3 The role of sparsity and comparison to Hybrid SINDy

We found that MODE using symbolic regressors (MODE-Symb) consistently outperformed a version with neural regressors (MODE-NN), suggesting a potential role for sparsity during learning. To investigate this phenomenon, we reran the Goldbeter experiment with MODE-Symb while setting the sparsity value to $\lambda = $ `0, 1e-4, 1e-3, 1e-2, 1e-1`. Each model was retrained with 5 random seeds.

We found that higher sparsity values encouraged more stable divisions of phase space into dynamical regimes across retrainings. This is visible if we plot the variance in cluster assignment averaged over all cells and across retrainings (Fig. 11) or view this variance across cells having retrained on different sparsity values. This complies with our intuition that less sparse (i.e more "complex" systems) should be able to fit the dynamics in multiple ways, and more sparse systems should converge to the same (in our experience also the true) expert distribution.

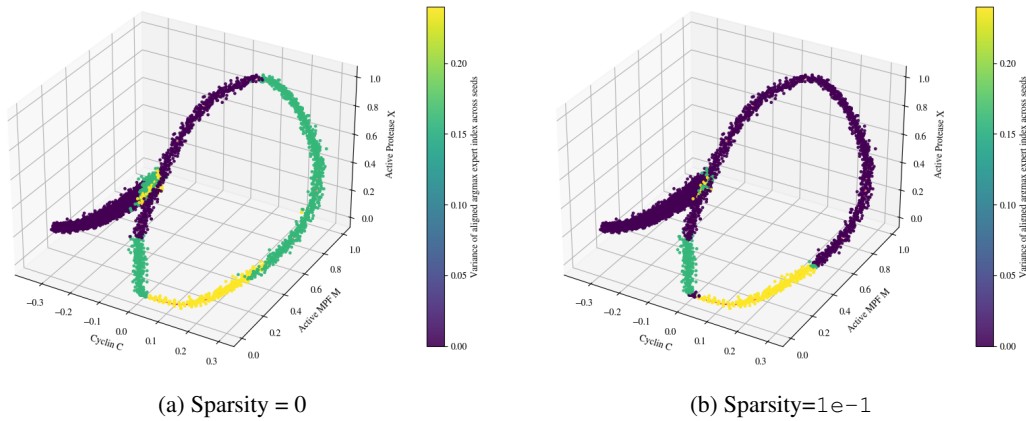

(a) Sparsity = 0         (b) Sparsity=`1e-1`

Figure 9: Variance in cluster assignment after MODE training with low vs high sparsity on Goldbeter.

This highlights an important difference between MODE and comparable switching/hybrid systems like Hybrid SINDy, where the regime assignments are learned before training. In our framework, model discovery and mixture modeling interact through learning. This means that we can use sparsity, for example, to help discover the true dynamical regimes. See, for example, the comparison between the Goldbeter regimes discovered by MODE vs Hybrid SINDy.

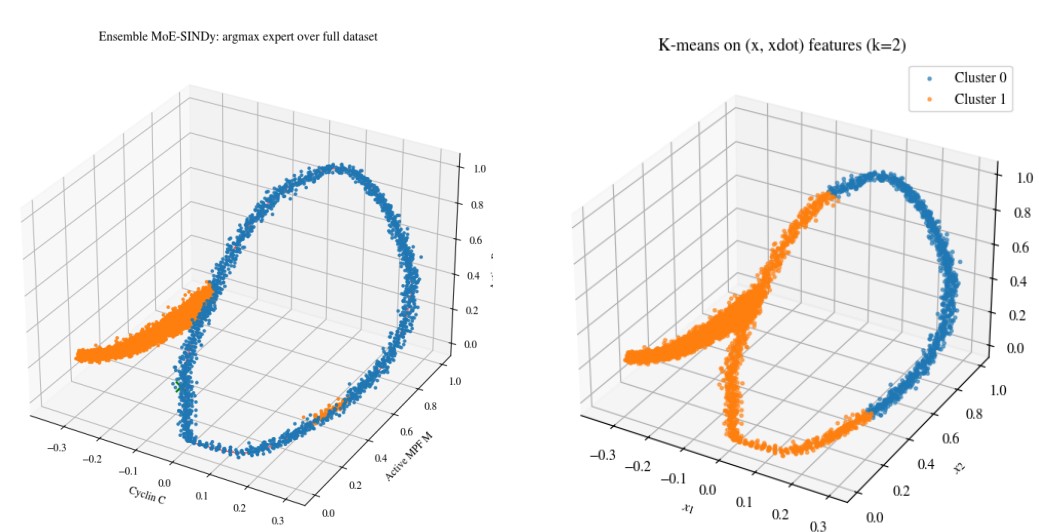

(a) Most likely cluster assignment according to ensemble of 5 MODEs with high sparsity.

(b) Most likely cluster assignment according to ensemble of 5 Hybrid SINDys with high sparsity.

Figure 10: Jointly learning regime assignments with dynamics using MODE correctly identifies true branches (MODE, left). Pre-clustering using Hybrid SINDy systematically leads to poor clustering.

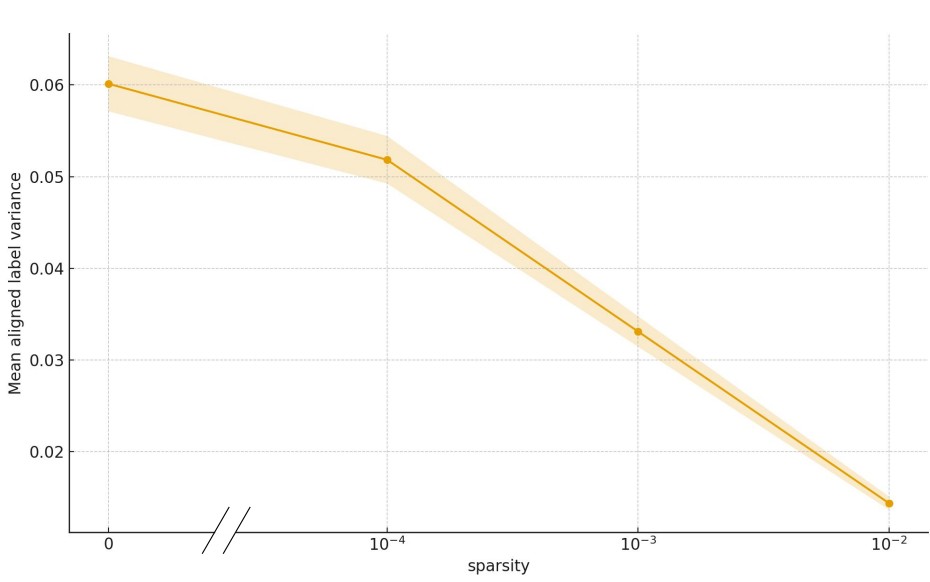

Figure 11: Variance of regime assignment with sparsity.

## B.4 SUPPLEMENTAL FIGURES

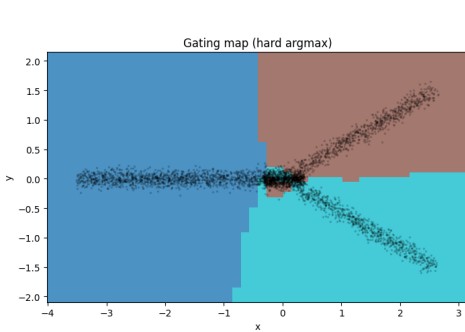
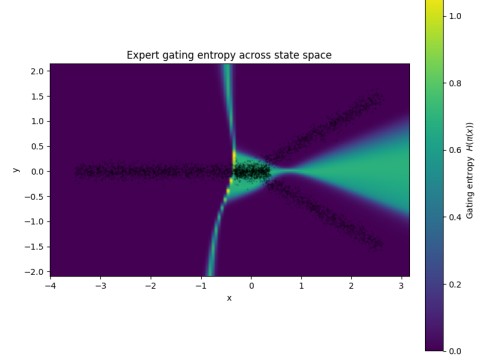

(a) Argmax of expert distribution learned for toy branching.

(b) Entropy of expert mixture, highlighting switching region.

Figure 12: Toy branching argmax and entropy figures

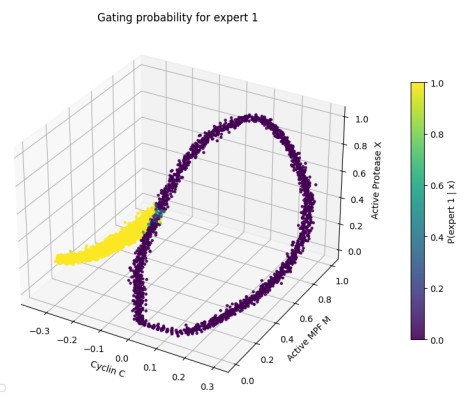
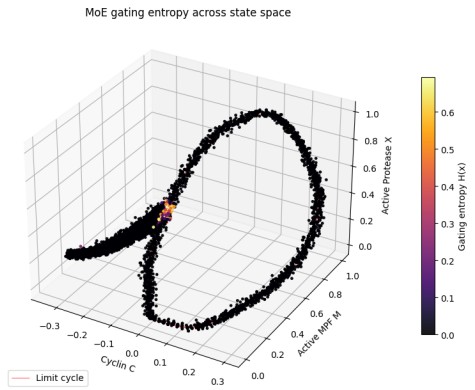

(a) Probability of differentiation expert (expert 1) learned for the Goldbeter system.

(b) Entropy of expert mixture, highlighting switching region.

Figure 13: Goldbeter probability and entropy figures

## B.5 OUT-OF-DISTRIBUTION (OOD) GENERALIZATION

To demonstrate MODE's ability to generalize to held-out data representing a large, contiguous portion of phase space, we ran an experiment in which a certain small arc on cell cycle was removed from the Goldbeter data. The sector was chosen to be away from the switching zone and near where protease starts being nonzero (red dots, Fig. 15, left). We found no difference (Fig. 15, right) in MODE's ability to detect the ground truth dynamical regimes on training (lightweight dots) vs OOD data (heavy blue dots). This indicates the bias introduced by MODE's sparse regressors helps "paint in" contiguous regions of phase space.

## B.6 CELL CYCLING IN HUMAN FIBROBLASTS

In Sec. 3 we used MODE in order to model experimental scRNA-seq data from the U2OS cell-line.

To further demonstrate the applicability of MODE on experimental data, we turned to data from Riba et al. of human fibroblasts. In this data, the cells can be separated into two subpopulations, only one of which exhibits genes related to the cell-cycle. A UMAP of this data can be seen in Fig. 16 (a), where cells are either part of the cell cycle or exiting it. We follow the same preprocessing steps as for the FUCCI data (Appendix Sec. D.1, estimate velocities using scVelo (Bergen et al., 2020),

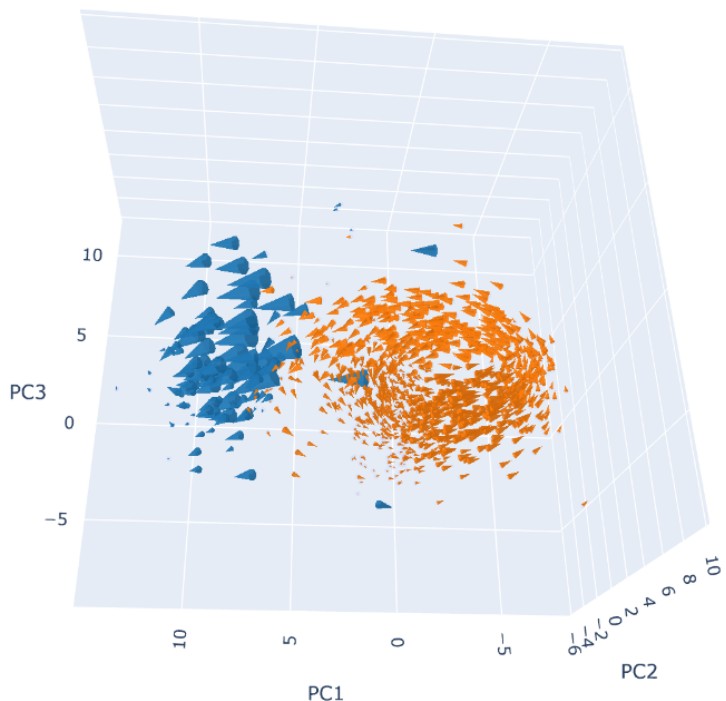

Figure 14: 3D plot of FUCCI cycle data with expert 1 and 2 in orange and blue respectively, highlighting the discovered cycling and exiting modes. Exiting cones doubled in size for visual emphasis.

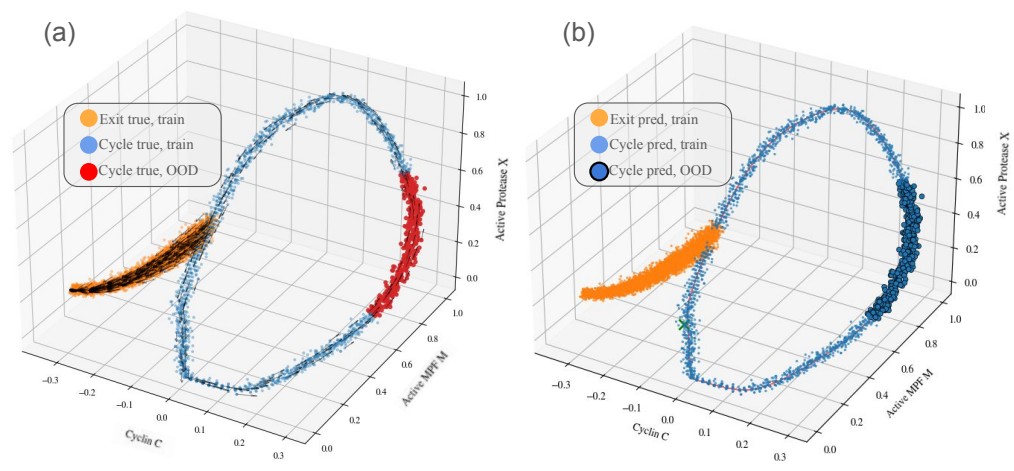

Figure 15: *MODE generalizes on Out-of-Distribution data.* (a) a 2-expert MODE was fit to the Goldbeter data with a sector of points hidden during training. (b) MODE is able to detect the correct dynamical regimes even in the region of space that was held-out during training.

which are then projected onto a 5-dimensional PCA. After the filtration steps, this data contains a total of 5,367 cells. Riba et al. showed it is possible to cluster these into cycling and non-cycling cells.

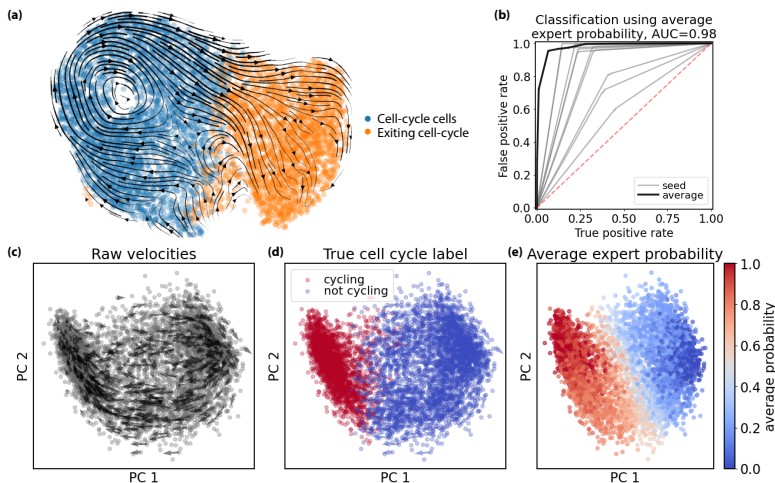

Figure 16: *Using MODE to model cell-cycle and exit dynamics in human fibroblasts. (a)* UMAP of the human fibroblasts data from Riba et al.. The cells can be divided into those that are part of the cell-cycle (in blue) and those that are not part of it (in orange). *(b)* MODE accurately distinguishes between these two regimes, and is stable between different seeds. *(c)* PCA projection of the data into two dimensions, with cell velocities. *(d)* true labeling into cell-cycle (blue) versus not (red). (e) classification according to 10 2-expert MODE, averaged over different seeds. The two different regimens, cycling and not, are captured by the two expert fit using MODE.

Following the results on U2OS, we fit this data with a 2-expert MODE, using up to quadratic terms, and repeated this process over ten random initializations to create an ensemble model (as described in Appendix Sec. D.1). We found that the ensemble averaged expert distribution is able to faithfully distinguish between cells that are cycling versus those that are not (Fig. 16 (b)). As in the U2OS data, the gating function had high entropy for cells between these regions, as shown in Fig. 16 (e).

## C  ALGORITHM IMPLEMENTATION DETAILS

We implement two variants of MODE corresponding to different assumptions about the mixing distribution $\pi$: MODE Local (state-independent mixing) and MODE Global (state-dependent mixing via neural gating).

### C.1  SPARSE IDENTIFICATION OF NONLINEAR DYNAMICS (SINDY)

The SINDy framework (Brunton et al., 2016) forms the foundation of our expert dynamics parameterization. Given snapshot data $D = \{(x_i, \dot{x}_i)\}_{i=1}^N$, SINDy assumes the governing equation can be expressed as a sparse linear combination of basis functions:

$$\dot{x} = f(x) = Z(x)\Theta,$$

where $Z(x) \in \mathbb{R}^{n_{\text{lib}}}$ is a library of candidate functions and $\Theta \in \mathbb{R}^{n_{\text{lib}} \times d}$ are sparse coefficients discovered via regularized regression. For polynomial libraries of degree $c$, we construct:

$$Z(x) = [1, x_1, \ldots, x_d, x_1^2, x_1 x_2, \ldots, x_d^c],$$

yielding $n_{\text{lib}} = \binom{d+c}{c}$ terms. The sparsity-promoting Laplace prior $\Theta_s \sim \text{Lap}(0, 1/\lambda)$ in Eq. 3 induces an $\ell_1$ penalty $\lambda\|\Theta\|_1$, implemented via Lasso regression. This enables automatic selection of relevant terms, yielding interpretable ordinary differential equations: each row of $\Theta$ specifies which polynomial terms contribute to each state derivative. For example, with $d = 2$ and $c = 2$, a discovered equation might be $\dot{x}_1 = \theta_2 x_2 + \theta_5 x_1 x_2$, directly revealing interaction structure.

In MODE Local (Sec. C), we solve weighted SINDy regressions in the M-step with responsibilities $R_{i,k}$ as sample weights. For numerical stability, we scale the design matrix and targets by $\sqrt{R_{i,k}}$, apply column-wise standardization without centering, fit Lasso, then rescale coefficients. In MODE Global (Sec. C.3), expert weight matrices $\Theta_k$ are directly optimized via gradient descent with $\ell_1$ regularization. The degree $c$ controls model complexity: higher degrees capture richer dynamics but increase overfitting risk.

We show in the clustering experiments that SINDy can accurately recover the true governing equations when they lie within the chosen basis (Tables 4,5). However, SINDy still works well, even when the true dynamics are not expressible in closed form in the basis. This is the case with the Goldbeter model, which uses rational functions. Polynomials should be an effective basis for this data away from singularities, especially when the degree of the basis functions grows larger. We can confirm this in a simple experiment in which the polynomial degree, $d$, is varied and the resulting model is evaluated using $W_1$ and $W_2$ pushforward losses (Fig. 17).

### C.2  MODE LOCAL

MODE Local implements the case where mixing probabilities $\pi = (\pi_1, \ldots, \pi_K)$ are state-independent constants. This variant is particularly suitable for unsupervised clustering of heterogeneous dynamical populations where each subpopulation follows distinct intrinsic dynamics regardless of spatial location.

**Algorithm.**  We optimize the MAP objective (Eq. 3) using an Expectation-Maximization (EM) algorithm with soft assignments. The algorithm alternates between:

**E-step:** Compute responsibilities $R_{i,k} = p(s_i = k \mid x_i, \dot{x}_i)$ for each data point $i$ and expert $k$:

$$R_{i,k} \propto \pi_k \cdot \mathcal{N}(\dot{x}_i \mid Z(x_i)\Theta_k, \sigma_k^2 I_d) \tag{17}$$

where responsibilities are normalized such that $\sum_{k=1}^K R_{i,k} = 1$.

**M-step:** Update expert parameters by solving weighted SINDy regression problems. For each expert $k$, we fit polynomial coefficients $\Theta_k$ by minimizing:

$$\sum_{i=1}^N R_{i,k} \|\dot{x}_i - Z(x_i)\Theta_k\|_2^2 + \lambda\|\Theta_k\|_1 \tag{18}$$

We also update mixing probabilities $\pi_k = \frac{1}{N}\sum_{i=1}^N R_{i,k}$ and noise variances $\sigma_k^2$.

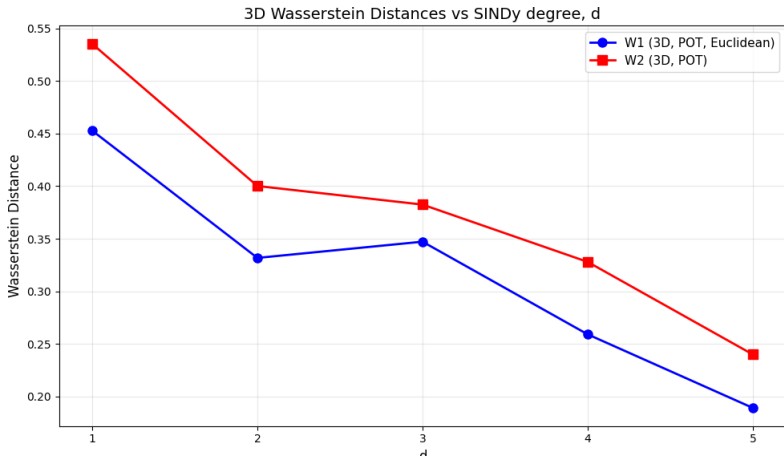

Figure 17: Degree, $d$, of the SINDy basis was varied while fitting Goldbeter data. We observed monotonically decreasing Wasserstein error as $d$ increased, indicating an increasingly accurate Taylor fit of the rational functions in the true dynamics.

**Implementation.** The core algorithm is implemented in the `MODELocal` class with the following key parameters:

- `n_clusters`: Number of experts $K$ (default: 3)
- `degree`: Polynomial basis degree for SINDy (default: 2)
- `alpha`: L1 regularization strength $\lambda$ (default: $10^{-4}$)
- `max_iter`: Maximum EM iterations (default: 150)
- `tol`: Convergence tolerance on log-likelihood (default: $10^{-5}$)

The weighted SINDy regression in the M-step uses sample re-weighting with stabilization: for each expert, we scale the design matrix $Z$ and targets $\dot{x}$ by $\sqrt{R_{i,k}}$, apply column-wise standardization without centering, fit Lasso regression, then rescale coefficients back to the original feature scale. This approach improves numerical stability compared to direct weighted least squares.

**Training Procedure.** The EM algorithm for MODE Local is summarized below:

The E-step computes soft assignments (responsibilities) $R_{i,k}$ for each data point based on the current expert parameters (lines 6–10). The M-step updates expert dynamics via weighted SINDy regression (lines 13–18), then updates mixing probabilities and noise variances (lines 21–22). Convergence is monitored via the marginal log-likelihood (line 25). The weighted regression uses sample re-weighting with column standardization (lines 14–17) to improve numerical stability.

### C.3 MODE GLOBAL

MODE Global implements the case where mixing probabilities depend on state: $\pi(x) = (\pi_1(x), \ldots, \pi_K(x))$. This variant uses a neural gating network to model spatially-localized regime transitions, making it suitable for forecasting tasks where dynamical behavior varies smoothly across state space.

**Algorithm.** We optimize Eq. 3 using stochastic gradient descent with the following neural architecture:

**Gating Network:** A 3-layer MLP with tanh activations maps normalized states to log-mixing probabilities:

$$\pi(x) = \text{softmax}\left(\text{MLP}\left(\frac{x - \mu_x}{\sigma_x}\right)\right) \tag{19}$$

---

**Algorithm 1** MODE Local Training (EM Algorithm)

---

**Require:** Data $(X, \dot{X})$, number of experts $K$, hyperparameters $\{\lambda, \text{degree}\}$
1: **Initialize** Expert parameters $\{\Theta_k\}_{k=1}^{K}$, mixing probabilities $\pi \in \mathbb{R}^K$, noise scales $\{\sigma_k\}_{k=1}^{K}$
2: Construct polynomial basis: $Z(X) \leftarrow \text{PolynomialFeatures}(X, \text{degree})$      $\triangleright$ (N, P) features
3:
4: **for** iteration $= 1$ to $N_{\text{iter}}$ **do**
5:
6:     *// E-step: Compute responsibilities*
7:     **for** $k = 1$ to $K$ **do**
8:        $v_k(X) \leftarrow Z(X)\Theta_k$        $\triangleright$ Expert predictions
9:        $\ell_{i,k} \leftarrow \log \pi_k + \log \mathcal{N}(\dot{x}_i | v_k(x_i), \sigma_k^2)$      $\triangleright$ Log-likelihood per sample
10:     **end for**
11:     $R_{i,k} \leftarrow \frac{\exp(\ell_{i,k})}{\sum_{j=1}^{K} \exp(\ell_{i,j})}$      $\triangleright$ Normalize responsibilities
12:
13:     *// M-step: Update parameters*
14:     **for** $k = 1$ to $K$ **do**
15:        *// Weighted SINDy regression*
16:        $w_i \leftarrow \sqrt{R_{i,k}}$        $\triangleright$ Sample weights
17:        $Z_w \leftarrow Z \odot w, \dot{X}_w \leftarrow \dot{X} \odot w$      $\triangleright$ Weight design matrix and targets
18:        Standardize columns of $Z_w$: $Z_w \leftarrow Z_w/\text{std}(Z_w)$
19:        Solve: $\Theta_k \leftarrow \arg\min_{\Theta} \|Z_w \Theta - \dot{X}_w\|_2^2 + \lambda \|\Theta\|_1$      $\triangleright$ Lasso regression
20:        Rescale $\Theta_k$ to original feature scale
21:
22:        *// Update mixing probability and noise*
23:        $\pi_k \leftarrow \frac{1}{N} \sum_{i=1}^{N} R_{i,k}$
24:        $\sigma_k^2 \leftarrow \frac{\sum_{i=1}^{N} R_{i,k} \|\dot{x}_i - v_k(x_i)\|_2^2}{\sum_{i=1}^{N} R_{i,k}}$
25:     **end for**
26:
27:     *// Check convergence*
28:     $\mathcal{L} \leftarrow \sum_{i=1}^{N} \log \sum_{k=1}^{K} \pi_k \mathcal{N}(\dot{x}_i | v_k(x_i), \sigma_k^2)$      $\triangleright$ Marginal log-likelihood
29:     **if** $|\mathcal{L} - \mathcal{L}_{\text{prev}}| < \text{tol}$ **then**
30:        **break**
31:     **end if**
32:     $\mathcal{L}_{\text{prev}} \leftarrow \mathcal{L}$
33: **end for**
34:
**Ensure:** Expert parameters $\{\Theta_k\}_{k=1}^{K}$, mixing probabilities $\pi$, noise scales $\{\sigma_k\}_{k=1}^{K}$

---

where $\mu_x, \sigma_x$ are empirical mean and standard deviation of the training data.

**Expert Networks:** Each expert $k$ parameterizes the dynamical law $f_{\Theta_k}(x) = Z(x)\Theta_k$ where $Z(x)$ contains polynomial features up to degree $c$ and $\Theta_k$ are learnable weight matrices of shape $(P, d)$ with $P$ polynomial features and $d$ output dimensions.

**Regularization.** To prevent mode collapse and ensure confident gating decisions, we augment the loss with:

- **Expert L1 sparsity:** $\lambda_{\text{expert}} \sum_{k=1}^{K} \|\Theta_k\|_1$ promotes sparse expert dynamics (applied only to linear SINDy experts)

- **Gate entropy:** $-\frac{1}{B} \sum_{i=1}^{B} \sum_{k=1}^{K} \pi_{i,k} \log \pi_{i,k}$ encourages confident per-sample decisions

- **Load balancing:** $\sum_{k=1}^{K} \bar{\pi}_k \log(K \bar{\pi}_k)$ prevents expert collapse via KL divergence from uniform distribution

  In practice, we find that a balance of entropy and load balancing terms works fairly well and is not too difficult to tune. Often, setting both to 1 produces fine results which can

be improved if necessary by either simple manual tuning or a grid search. The example of a gridsearch on gating entropy and load balancing in which Wasserstein 1-distance is measured on Goldbeter testing data is given in Fig. 18.

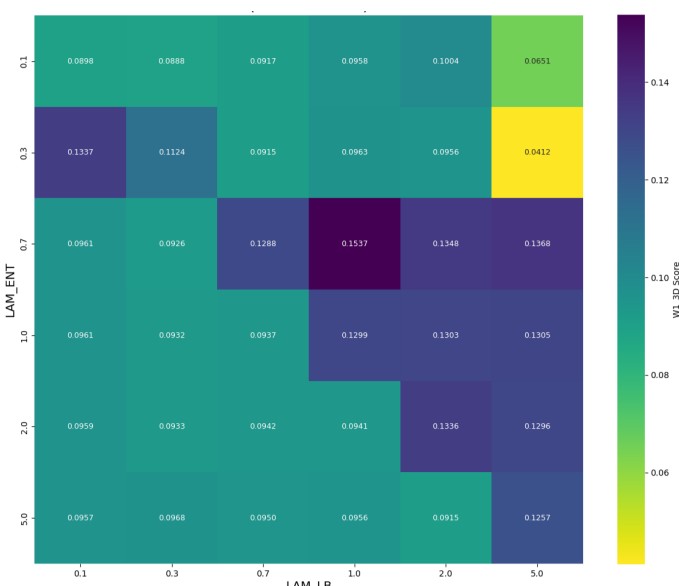

Figure 18: Hyperparameter grid search on the Goldbeter data.

**Rollouts** All results in the paper use the stochastic rollout described in Eqs. 5. Alternatively, we investigated using "mean" and "max" rollouts in which the average or mode of the expert distribution is used at every step, respectively. This represents a different type of cellular dynamics in which regulatory processes either have a conjoined, simultaneous effect or a winner-takes-all dynamics. Table 7 shows Wasserstein scores on the Goldbeter data for these different dynamics.

|  | $W_1$ | $W_2$ | $W_{1,C}$ | $W_{1,M}$ | $W_{1,X}$ |
|---|---|---|---|---|---|
| Max | 0.096034 | 0.152445 | **0.022993** | 0.039216 | 0.050021 |
| Mean | 0.101588 | 0.160200 | 0.024213 | 0.041055 | 0.055026 |
| Stochastic | **0.0837** | **0.1049** | 0.0590 | **0.0223** | **0.0289** |

Table 7: Wasserstein distances for different MODE rollout methods. Stochastic rollouts perform best on this Goldbeter data.

**Implementation.** The algorithm is implemented in the `MODEGlobal` class with two training modes: fixed-K and adaptive-K. For all clustering experiments (Sec. 3.1), the key hyperparameters were set to:

- `K`: Number of experts (fixed mode: default 2)
- `degree`: Polynomial degree for expert dynamics (default: 2)
- `hidden`: Hidden units in gating network (default: 64)
- `epochs`: Training epochs (default: 1500 )
- `lr`: Learning rate (default: $1 \times 10^{-3}$)
- `l1_lambda`, `ent_lambda`, `lb_lambda`: Regularization weights (defaults: $10^{-1}, 1, 1$)

MODE training and hyperparameters for all experiments are shown in tables Tab. 8, 9 and 11. Wherever parameters are dashed, they were omitted or set to 0 as they had no influence on the result.

All MLP baselines had four hidden layers (except for the two data which had only one layer) with 64 units in each layer and `tanh` activation functions.

| Parameter | Description | Value |
|---|---|---|
| K | experts | 3 |
| LIB_ORDER | poly. degree | 0 |
| GATE_HIDDEN | gating hidden sizes | (64,) |
| ACTIVATION | nonlinearity | tanh |
| LR | learning rate | 2e-3 |
| WEIGHT_DECAY | L2 penalty | 1e-5 |
| EPOCHS | training epochs | 500 |
| PATIENCE | early stop patience | 30 |
| MIN_DELTA | val. loss margin | — |
| LAM_L1 | L1 penalty | 1e-4 |
| LAM_ENT | entropy penalty | 1e-3 |
| LAM_LB | load balance penalty | 5e-4 |
| GRAD_CLIP | gradient clipping | — |
| BATCH_SIZE | batch size | 512 |

Table 8: Toy Branching (i.e. Fig. 1) hyperparameters.

| Parameter | Description | Value |
|---|---|---|
| K | experts | 2 |
| LIB_ORDER | poly. degree | 3 |
| GATE_HIDDEN | gating hidden sizes | (64,) |
| ACTIVATION | nonlinearity | SILU |
| LR | learning rate | 1e-2 |
| WEIGHT_DECAY | L2 penalty | 1e-6 |
| EPOCHS | training epochs | 10000 |
| PATIENCE | early stop patience | 30 |
| MIN_DELTA | val. loss margin | 1e-4 |
| LAM_L1 | L1 penalty | 1e-3 |
| LAM_ENT | entropy penalty | 1e0 |
| LAM_LB | load balance penalty | 1e0 |
| GRAD_CLIP | gradient clipping | 5.0 |
| BATCH_SIZE | batch size | 512 |

Table 9: Goldbeter oscillator hyperparameters.

All implementations support multiple random restarts with validation-based model selection to avoid poor local minima. For reproducibility, all experiments use fixed random seeds across method comparisons. GMM and spectral clustering baselines were run in `scikit-learn v. 1.7.2`. MODE, MLPs and SINDy were implemented in `pytorch v. 2.8.0` and were trained with an Adam optimizer (Kingma & Ba, 2015). Experiments were run locally on a cpu cluster with no parallelization. Typical run times range from around a minute for the toy example of Fig. 1 to about ten minutes for the full fifteen-seed FUCCI ensemble shown in Fig. 5 *(e)*, *(f)*.

**Training Procedure.** The complete training algorithm is summarized below:

The forward pass computes gating probabilities $\pi(x)$ and expert predictions $\{v_k(x)\}$ (lines 7–8), which are combined via a Gaussian mixture likelihood (line 11). Regularization terms (lines 14–17) prevent mode collapse and encourage sparse, confident expert assignment. The total loss is minimized using Adam optimizer (lines 20–21).

## C.4 Adaptive Expert Selection

In many biological systems, the number of distinct dynamical regimes is *a priori* unknown. While cell population heterogeneity is ubiquitous in single-cell data, determining how many functionally

| Parameter | Description | Value |
|---|---|---|
| K | experts | 3 |
| LIB_ORDER | poly. degree | 1 |
| GATE_HIDDEN | gating hidden sizes | (64,) |
| ACTIVATION | nonlinearity | tanh |
| LR | learning rate | 1e-2 |
| WEIGHT_DECAY | L2 penalty | 1e-4 |
| EPOCHS | training epochs | 2000 |
| PATIENCE | early stop patience | 30 |
| MIN_DELTA | val. loss margin | 1e-4 |
| LAM_L1 | L1 penalty | 1e-3 |
| LAM_ENT | entropy penalty | 2e-3 |
| LAM_LB | load balance penalty | 1e-1 |
| GRAD_CLIP | gradient clipping | 5.0 |
| BATCH_SIZE | batch size | 512 |

Table 10: Lineage branching hyperparameters.

| Parameter | Description | Value |
|---|---|---|
| K | experts | 2 |
| LIB_ORDER | poly. degree | 3 |
| GATE_HIDDEN | gating hidden sizes | (64,) |
| ACTIVATION | nonlinearity | SILU |
| LR | learning rate | 1e-3 |
| WEIGHT_DECAY | L2 penalty | 1e-6 |
| EPOCHS | training epochs | 5000 |
| PATIENCE | early stop patience | 100 |
| MIN_DELTA | val. loss margin | 1e-4 |
| LAM_L1 | L1 penalty | 1e-3 |
| LAM_ENT | entropy penalty | 1e0 |
| LAM_LB | load balance penalty | 1e0 |
| GRAD_CLIP | gradient clipping | 5.0 |
| BATCH_SIZE | batch size | 512 |

Table 11: FUCCI hyperparameters.

distinct subpopulations exist - and whether these correspond to discrete dynamical regimes or continuous transitions - remains a fundamental challenge. Manual selection of the expert count $K$ is both impractical and prone to overfitting or underfitting. We therefore introduce an AIC/BIC-based strategy for dynamically adapting $K$ during training, allowing the model to automatically discover the intrinsic dimensionality of the system.

Starting with an initial $K$ (default: 5), the algorithm periodically evaluates information criteria to balance model complexity with goodness-of-fit:

$$\text{AIC} = 2k + 2N \cdot \text{NLL} \tag{20}$$
$$\text{BIC} = k \log(N) + 2N \cdot \text{NLL} \tag{21}$$

where $k$ is the number of model parameters, $N$ is the number of samples, and NLL is the negative log-likelihood. The adaptation rules are:

- **Add expert:** If AIC/BIC plateaus for $P_{\text{add}}$ consecutive checks (default: 30 checks $\times$ 50 epochs = 1500 epochs), suggesting the model needs more capacity to improve fit.
- **Remove expert:** If (1) any expert has usage $< \tau_{\text{usage}}$ (default: 5%), indicating redundancy, or (2) AIC/BIC worsens consistently for $P_{\text{remove}}$ checks (default: 30), suggesting overfitting. The least-used expert is removed.

To further encourage sparsity in expert assignment, we add the regularization term $\lambda_{\text{gate}} \sum_{p \in \text{gating}} |p|$ to the loss, which penalizes gating network weights and prevents diffuse, non-interpretable expert usage (default: $\lambda_{\text{gate}} = 10^{-3}$).

---

**Algorithm 2** MODE Global Training

---

**Require:** Data $(X, \dot{X})$, number of experts $K$, hyperparameters $\{\lambda_{\text{expert}}, \lambda_{\text{ent}}, \lambda_{\text{lb}}\}$
 1: **Initialize** Gating network MLP, expert parameters $\{\Theta_k\}_{k=1}^K$, noise scales $\{\sigma_k\}_{k=1}^K$
 2: Normalize data: $\mu_x \leftarrow \text{mean}(X)$, $\sigma_x \leftarrow \text{std}(X)$
 3: **for** epoch $= 1$ to $N_{\text{epochs}}$ **do**
 4:     **for** minibatch $(x_b, \dot{x}_b)$ in DataLoader$(X, \dot{X})$ **do**
 5:        *// Forward pass*
 6:        $\pi(x_b) \leftarrow \text{softmax}(\text{MLP}((x_b - \mu_x)/\sigma_x))$        ▷ Gating probabilities
 7:        $\{v_k(x_b)\}_{k=1}^K \leftarrow \{Z(x_b)\Theta_k\}_{k=1}^K$        ▷ Expert predictions
 8:        *// Mixture likelihood*
 9:        $\mathcal{L}_{\text{data}} \leftarrow -\frac{1}{B} \sum_{i=1}^B \log \sum_{k=1}^K \pi_k(x_i) \cdot \mathcal{N}(\dot{x}_i | v_k(x_i), \sigma_k^2)$
10:        *// Regularization*
11:        $\mathcal{L}_{\text{expert}} \leftarrow \lambda_{\text{expert}} \sum_{k=1}^K \|\Theta_k\|_1$        ▷ Expert sparsity
12:        $\mathcal{L}_{\text{ent}} \leftarrow -\lambda_{\text{ent}} \cdot \frac{1}{B} \sum_{i=1}^B \sum_{k=1}^K \pi_{i,k} \log \pi_{i,k}$        ▷ Gate entropy
13:        $\bar{\pi} \leftarrow \frac{1}{B} \sum_{i=1}^B \pi_i$        ▷ Average expert usage
14:        $\mathcal{L}_{\text{lb}} \leftarrow \lambda_{\text{lb}} \sum_{k=1}^K \bar{\pi}_k \log(K \bar{\pi}_k)$        ▷ Load balancing
15:        *// Update*
16:        $\mathcal{L}_{\text{total}} \leftarrow \mathcal{L}_{\text{data}} + \mathcal{L}_{\text{expert}} + \mathcal{L}_{\text{ent}} + \mathcal{L}_{\text{lb}}$
17:        Backpropagate $\nabla \mathcal{L}_{\text{total}}$ and update parameters with Adam
18:     **end for**
19:     **if** early stopping criterion met **then**
20:        **break**
21:     **end if**
22: **end for**
**Ensure:** Trained gating network and expert parameters

---

Figure 5$(e), (f)$ demonstrates the robustness of this adaptive strategy across 15 independent training runs with different random initializations. Despite starting from $K = 5$ experts, all runs converge to $K = 2$ experts, with the mean trajectory (purple line) showing rapid initial adaptation followed by stable convergence. The right panel shows that AIC values decrease sharply during adaptation and stabilize once the optimal expert count is reached, with all runs identifying similar final model complexities (red dots mark minimum AIC). The algorithm maintains $K_{\min} \leq K \leq K_{\max}$ (default: $2 \leq K \leq 10$) and resets patience counters after each adaptation to allow the model to stabilize. This consistency across random seeds confirms that the method reliably discovers the intrinsic dimensionality of the dynamical regimes without requiring prior knowledge of the optimal $K$.

**Note on experimental results.** While this adaptive selection capability is critical for exploratory analyses where the number of regimes is unknown, *the main results presented in this paper were obtained with fixed $K$ values*, as the number of biologically relevant populations was known *a priori* from experimental annotations. The adaptive strategy is provided as a tool for discovery-driven applications and to demonstrate the model's robustness to initialization, but was not required for the validation experiments described in Section 3.

## D BIOLOGICAL DATASET DETAILS

### D.1 FUCCI DATASET

The FUCCI (Fluorescent Ubiquitination-based Cell Cycle Indicator) dataset used in this study originates from the comprehensive single-cell proteogenomics analysis conducted by Mahdessian et al. (2021). This dataset represents a unique integration of single-cell RNA sequencing with fluorescent cell cycle markers in the U2OS human osteosarcoma cell line, providing ground truth annotations for cell cycle phase identification.

**Dataset composition and cell cycle labeling.** The original dataset comprises 1,152 individual cells characterized by the expression profiles of 58,884 genes. The FUCCI system employs two

fluorescently tagged cell cycle markers: CDT1 (tagged with RFP, expressed during G1 phase) and GMNN (tagged with GFP, expressed during S and G2 phases). Cells expressing both markers simultaneously indicate the G1-S transition phase. This dual-marker system provides precise temporal information about cell cycle progression, enabling the identification of cells in cycling versus non-cycling (differentiation) states. The FUCCI markers serve as our ground truth labels for distinguishing between two fundamental dynamical regimes: (1) active cell cycle progression and (2) cell cycle exit leading to differentiation.

**Preprocessing pipeline.** We followed the preprocessing protocol established by Zheng et al. (2023), which includes several critical steps for robust scRNA-seq analysis:

- **Gene filtering:** Removal of genes with low expression counts across the cell population to reduce noise and computational burden.
- **Cell filtering:** Exclusion of cells with abnormally low or high gene counts that might indicate technical artifacts or doublets.
- **Normalization:** Application of library size normalization to account for differences in sequencing depth between cells.
- **Log transformation:** $\text{Log}_{10}(\text{count} + 1)$ transformation to stabilize variance and reduce the influence of highly expressed genes.
- **Highly variable gene selection:** Identification and retention of genes showing significant cell-to-cell variability, which are most informative for capturing biological differences.

**RNA velocity estimation.** We employed scVelo (Bergen et al., 2020) to estimate RNA velocity vectors from the preprocessed data. scVelo computes velocity by modeling the dynamics of gene expression through an analytical framework that considers transcription, splicing, and degradation rates. Specifically, the method uses the ratio of unspliced to spliced mRNA counts to infer the directional flow of gene expression changes. This provides local velocity estimates $\dot{x}_i$ for each cell $i$, representing the instantaneous rate of change in gene expression space. The velocity estimation captures the intrinsic directionality of cellular state transitions, making it particularly suitable for identifying developmental trajectories and regime transitions.

**Dimensionality reduction and feature space.** Given the high-dimensional nature of the gene expression data (58,884 dimensions), we applied Principal Component Analysis (PCA) to project both the gene expression profiles and their corresponding velocity vectors into a 5-dimensional latent space. This dimensionality reduction serves multiple purposes: (1) computational efficiency for MODE training, (2) noise reduction by focusing on the most informative directions of variation, and (3) visualization feasibility for the first three principal components. The choice of 5 dimensions balances the preservation of essential biological signals while maintaining computational tractability for the mixture of experts framework.

**MODE training configuration.** Full hyperparameters in Tab. 11. An ensemble model created by averaging (e.g. over mixture distributions and expert parameters; experts were aligned by similarity of parameter vectors) over 10 random seeds. All seeds but one (the outlier curve in Fig. 5) divided the data into clear cycle and exit regimes.

**Validation methodology.** Model performance was evaluated using the FUCCI ground truth labels as the gold standard for cell cycle phase classification. We computed Receiver Operating Characteristic (ROC) curves for each of the 10 model initializations, measuring the ability of the learned expert probabilities to distinguish cycling from differentiating cells. The consistently high Area Under the Curve (AUC) scores ($0.98 \pm 0.01$) across all initializations demonstrate both the reliability of our approach and the biological relevance of the discovered dynamical regimes. Additionally, we validated the biological interpretability of the learned dynamics by examining the correspondence between expert assignments and known cell cycle markers, confirming that the two experts successfully captured the cycling and differentiation programs respectively.

**Transcriptional and regulatory validation.** Beyond phase classification accuracy, we validated whether MODE's learned dynamics capture biologically interpretable structure through multilevel

analysis of gene expression signatures, temporal dynamics, and regulatory network topology. Importantly, MODE was trained without any prior knowledge of cell cycle markers or temporal annotations—all biological interpretations were performed post-hoc.

EXPERT-SPECIFIC DIFFERENTIAL EXPRESSION METHODOLOGY. We compared gene expression between cells assigned to Expert 0 versus Expert 1 using dominant expert assignments as the basis for grouping. For each gene $g$, we computed mean expression in each expert group and assessed differential expression using fold-change $\text{FC}_g = \log_2(\bar{x}_g^{(1)}/\bar{x}_g^{(0)})$ with statistical significance via permutation tests ( FDR-corrected $p < 0.001$).

Remarkably, without supervision, Expert 0 spontaneously enriched for canonical proliferation markers (TOP2A, CDK1, AURKA) and chromatin organization factors (HIST1H1C, HIST1H4C), while Expert 1 enriched for known cell cycle exit-associated genes (CAV1, THBS1, CCN1, CDKN1A). Comparison with established marker gene sets from the literature (Malumbres & Barbacid, 2009; Coller et al., 2006) confirmed that MODE's unsupervised expert discovery recovered biologically meaningful cell states (Table 12).

| Upregulated in Expert 1 | | | Downregulated in Expert 1 | | |
|---|---|---|---|---|---|
| **Gene** | **Role** | **Change** | **Gene** | **Role** | **Change** |
| CAV1 | Quiescence | 28.0 | KRT17 | Proliferation | -41.0 |
| CDC20 | APC/C activator | 26.2 | TGFBI | ECM protein | -11.1 |
| THBS1 | Anti-angiogenic | 23.7 | HIST1H1C | Chromatin | -10.2 |
| TPX2 | Mitotic spindle | 21.2 | TOP2A | DNA topology | -7.8 |
| CCNB1 | G2/M cyclin | 20.5 | CDK1 | Cell cycle kinase | -5.7 |

Table 12: Top differentially expressed genes between experts

PSEUDOTEMPORAL ORDERING AND TEMPORAL DYNAMICS. We constructed a pseudotime trajectory by sorting cells according to Expert 0 probability $\pi_0(x_i)$, naturally ordering cells from high $\pi_0$ to low $\pi_0$ states. Gene expression profiles were smoothed using Gaussian kernel smoothing ($\sigma = 50$ cells), and the transition point was identified as the pseudotime value maximizing expert entropy: $t^* = \arg\max_t H(\pi(x_t))$, where $H(\pi) = -\sum_k \pi_k \log \pi_k$.

Along this trajectory, Expert 0-enriched markers (TOP2A, CDK1) decreased while Expert 1-enriched markers (CAV1, THBS1) increased, with the steepest changes occurring near $t^*$ (Fig. 6A). Expression profiles showed gradual downregulation of proliferation-associated genes preceding upregulation of exit-associated genes, suggesting sequential transcriptional reprogramming consistent with established models of cell cycle exit (Spencer et al., 2013).

GENE REGULATORY NETWORK EXTRACTION AND REWIRING ANALYSIS. For linear SINDy experts with dynamics $\dot{x} = Z(x)\Theta_k$, the local Jacobian in PCA space is computed from the learned coefficient matrices $\Theta_k$ and the basis family $Z$. To map Jacobians to the original gene space, we applied the chain rule through PCA projection $x_{\text{PCA}} = V(x_{\text{gene}} - \mu)$, where $V \in \mathbb{R}^{d_{\text{PCA}} \times d_{\text{gene}}}$ are the PCA components:

$$J_k^{\text{gene}} = V^\top J_k^{\text{PCA}} V$$

We extracted inferred gene regulatory networks (GRNs) for three cell state regions based on expert dominance: before transition (Expert 0 probability $> 0.8$), during transition (expert entropy in top 20%), and after transition (Expert 1 probability $> 0.8$). For each region, we computed mean Jacobians over sampled cells ($n = 50$ per region) and applied a sparsity threshold of $|\tilde{J}_{ij}| > 0.15$ after normalization to retain the strongest interactions.

Network topology analysis revealed substantial rewiring across transition phases (Fig. 6B). Before transition, the inferred network featured prominent interactions among proliferative genes including the CDK1–CCNB1–CDC20 module. After transition, exit-associated genes (CAV1, THBS1) gained regulatory prominence while proliferative gene interactions weakened. These patterns of network reorganization are consistent with known regulatory circuitry governing cell cycle progression and exit (Malumbres & Barbacid, 2009; Pines, 1995).

HIGH-ENTROPY TRANSITION ANALYSIS. To identify genes specifically associated with the transition state, we compared gene expression between cells in high-entropy regions (top 20% by gating network entropy: $H(\pi) > \theta_H$, where $\theta_H = \text{percentile}_{80}(H)$) versus low-entropy cells (bottom 20%). This identified 222 genes with significant expression changes ($|FC| > 0.5$, permutation $p < 0.001$, Fig. 6C).

Notably, top transition-associated genes (including CDC20B, PDK4, and RASSF6) differed from the markers defining stable expert states (e.g., TOP2A for Expert 0, CAV1 for Expert 1), suggesting these genes represent transient regulatory factors active during the decision phase rather than stable markers of committed states. This demonstrates MODE's ability to distinguish stable dynamical regimes from transitional cell states.

VALIDATION SUMMARY. Collectively, these analyses demonstrate that MODE's unsupervised learning recovers biologically coherent structure: (1) experts spontaneously align with known cell cycle states without prior annotation, (2) pseudotemporal ordering reveals sequential transcriptional dynamics matching established exit mechanisms (Spencer et al., 2013), (3) inferred regulatory networks exhibit topology changes consistent with known cell cycle circuitry (Malumbres & Barbacid, 2009; Pines, 1995), and (4) transition-state cells show distinct gene expression profiles from stable states. Together, this multilevel concordance with domain knowledge validates that MODE captures mechanistically meaningful dynamics rather than arbitrary data partitions.

## SUPPLEMENTAL VIDEOS

We have also included with this submission four videos showing MLP vs MODE on toy branching data, the stochastic MODE rollout on the Goldbeter oscillator as well as the rollout for our fit of the FUCCI RNA velocity data.

## REPRODUCIBILITY STATEMENT

All details necessary to reproduce the results of this work can be found in the appendix and supplementary materials:

- Complete specifications of the MODE architecture, including network architectures, loss functions, and training procedures are provided in Appendix C.

- Hyperparameter choices for all experiments, including learning rates, regularization weights, and model selection criteria can be found in Appendix C.

- Governing equations and parameter values for all synthetic dynamical systems are described in Appendix A for the elementary dynamics benchmarks and Appendix A.2 for the forecasting experiments.

- Detailed preprocessing steps for the FUCCI biological dataset, including gene filtering criteria, normalization procedures, and dimensionality reduction parameters are given in Appendix D.1.

- Source code implementing the MODE framework, including both EM-based and neural-gated variants, is available in the `notebooks/` directory with documented examples reproducing all main results.

- All synthetic datasets can be regenerated using the equations and parameters provided in the appendices, while the FUCCI dataset is publicly available from Mahdessian et al. (2021).

- Computational requirements and runtime specifications are documented for all experiments, with typical runtimes.

We have made every effort to ensure that our experimental methodology is transparent and our results are fully reproducible by the research community. Moreover, we provide our complete implementation at `https://github.com/anonresearcher22-netizen/MODE`.

## LARGE LANGUAGE MODEL USAGE STATEMENT

In accordance with ICLR 2026 policies on Large Language Model usage, we disclose that Large Language Models (specifically Claude Sonnet 3.5) were used to assist in the preparation of this manuscript in the following ways:

- **Writing assistance**: LLMs were used to typeset equations and format tables in LaTeX. It was also used to summarize implementational details from code into sections of the Appendix.

- **Literature review support**: LLMs assisted in identifying relevant references and helped structure the related work section, though all cited works were independently verified and evaluated by the authors.

- **Code documentation and presentation**: LLMs were employed to enhance code comments, improve code readability, and assist in the preparation of well-structured, documented implementations in the supplementary materials.

We emphasize that all core research contributions, experimental design, data analysis, and scientific conclusions are entirely the work of the human authors. The authors take full responsibility for the accuracy and validity of all content, including any text that may have been refined with LLM assistance. All LLM-generated content was carefully reviewed, fact-checked, and validated by the authors before inclusion. No LLMs were used for generating experimental results, conducting data analysis, or making scientific claims.

