# OpenReview forum: "MODE: Learning compositional representations of complex systems with Mixtures Of Dynamical Experts"
_ICLR.cc/2026/Conference — Submitted to ICLR 2026_

### Official Review · Reviewer_1SNf · 2025-10-26

**Soundness:** 3
**Presentation:** 3
**Contribution:** 2
**Rating:** 4
**Confidence:** 3

**Summary:**

The authors propose a unified perspective to address key limitations of existing methods for modelling complex mixtures of overlapping behavioural regimes: the inability of flow-based models to handle branching dynamics, the computational complexity of switching system inference, and the lack of interpretability in many neural approaches. The resulting Mixture Of Dynamical Experts (MODE) framework shows improved performance over baselines like Gaussian Mixture Models, Neural ODEs, etc. across a range of idealised and real-world benchmarks.

**Strengths:**

1. The paper clearly identifies the failure of traditional flow-based models (like NODEs) in handling complex, overlapping dynamical regimes, using Figure 1 effectively to show how these models improperly "average" distinct fates.
2. The MODE framework is great for its simplicity, combining an intuitive Mixture of Experts (MoE) approach with SINDy-based regressors. This grounds the model in interpretable, sparse symbolic equations, which is a significant advantage for scientific applications.
3. The "Related Work" section is thorough, correctly positioning the paper by contrasting it against standard dynamical systems, specific computational biology flow models, switching systems, and other MoE approaches.

**Weaknesses:**

I am not an expert in computational biology, so here are the weaknesses I underline, mostly relating to the machine learning methodology:

1. A priori $K$ selection: The most critical limitation of this work is that the number of experts, $K$, must be manually specified corresponding to the number of dynamic regimes. This is impractical for real-world discovery tasks where $K$ is a primary unknown.
2. The paper's solution to trajectory crossing (a known failure of NODEs) is clear. It would be strengthened by contrasting its discrete mixture-based approach with continuous, augmentation-based methods like ANODE [1].
3. Gradient-based optimisation of the gating network is a known challenge (see MixER [2]). The paper's regularization solution (Eq. 4) to prevent the gating network from collapsing is interesting. It could be compared to other methods, such as the K-Means initialization used in MixER.
4. A claim in L297-298 that MODE "does not rely on phase space geometry" appears to contradict the model's formulation, which explicitly uses the state $x$ (the geometry) to parameterize both the gating function $\pi(x)$ and the expert dynamics $f_{\Theta_{s}}(x)$. This needs clarification.



### Minor issues:
- L235: "dynamical"
- L236: "on"

### References:
- [ 1 ] Dupont et al., "Augmented Neural ODEs", NeurIPS 2019
- [ 2 ] Nzoyem et al., "MixER: Better Mixture of Experts Routing for Hierarchical Meta-Learning", SCOPE@ICLR 2025.

**Questions:**

1) Is the claim that NODEs "average" switching zones (L067) an empirical observation of a common training failure, or a more fundamental theoretical limitation of fitting a single vector field to multi-modal velocity data? Please compare with ANODE for instance?
2) There is minimal quantitative comparison to Neural ODEs, even though there are the subject of much of the criticism in the Related Work. This relates to my question in 1)
3) Eq (4) displays two losses that aim to achieve two opposite things. I have two questions concerning this:
	- It is not clear how the load balancing term prevents expert collapse (despite the additional definition in L1055)
	- Could you please provide an ablation for this?
4) Regarding ablation studies, one examining the polynomial basis of SINDy is much needed, especially when the oscillator uses functions outside the dictionary (Goldbeter experiment).

---

> ### Author Response · Authors · 2025-11-21
>
> We thank the reviewer for these constructive remarks, particularly as they touch on our model’s hyperparameter settings. We address each remark in turn:
>
>
> - “The most critical limitation of this work is that the number of experts, , must be manually specified”.
> To address this concern, we have now implemented a new algorithm in which the number of experts can be adaptively selected during learning based on standard information-theoretic criteria like the Akaike or Bayes Information Criteria. In a new experiment (App. Sec. C “Adaptive Expert Selection”) we show how this algorithm then consistently converges to  K=2 experts on the FUCCI data across 15 reruns. The criteria are tried-and-true in the model discovery literature and we find they interface easily with our approach, showing that lack of prior knowledge on the number of experts is not a limitation of our method.
>
> - “Is the claim that NODEs "average" switching zones (L067) an empirical observation of a common training failure, or a more fundamental theoretical limitation of fitting a single vector field to multi-modal velocity data?”
> We view the switching zones as a fundamental limitation that arises in dynamical data with crossing trajectories. The reviewer’s suggestion to compare to ANODEs is intriguing, as these too could model crossing trajectories. For the moment, we are primarily interested in stochastic systems where crossing trajectories are caused by stochastic changes in the parameters in the underlying circuits as opposed to changes in a hidden, latent variable. We now include a brief discussion of ANODEs and their relation to our work in the discussion section.
>
> - “There is minimal quantitative comparison to Neural ODEs, ” We have now added two new baselines, one of which, MODE-NN (Table 2, Row 4), is a version of our framework in which the sparse regressors are replaced with NODEs. We show how this ablated (in the sense of having no sparsity) model performs the closest to our proposal, indicating an advantage earned by mixture modeling, but a disadvantage in its lack of sparsity. This sparsity effect is further explored in App. Sec. C (“The Role of Sparsity”).
>
> - “Eq (4) displays two losses that aim to achieve two opposite things. I have two questions concerning this:”. We found that the load balancing term penalizes scenarios where one expert receives most assignments across the whole data set, ensuring all experts are used equally across the batch and thus directly preventing expert collapse. A new experiment in which the two regularizers are systematically varied is found in Fig. 17 of the appendix. This shows a nontrivial interaction between these two regularizers on the Goldbeter data.
>
>
> - “Regarding ablation studies, one examining the polynomial basis of SINDy is much needed” Thank you for this suggestion. We carried out this experiment in Fig. 16, which shows, somewhat intuitively, that increasing the polynomial degree in approximating the rational functions underlying Goldbeter leads to strictly better performance.

---

> > ### Comment · Reviewer_1SNf · 2025-11-26
> >
> > Dear Authors, I thank you for your rebuttal efforts.
> > - This new experiment sounds interesting. I framed this as a fundamental limitation, and the fact that it is buried in the Appendix would hide its value from the ICLR community.
> > - Thanks for incorporating ANODE.
> > - Thanks for these as well. I wonder, if incorporating NODE was possible, why wasn't your sparse regressors are replaced with ANODE as well?
> > - Figure 17 appears to be captioned "Hyperparameter grid search on the Goldbeter data". I am not sure how it helps support your point or addresses any of the two questions. Could you please clarify ?
> > - This matches my intuition as well. Thank you for confirming this empirically.

---

> > > ### Author Response · Authors · 2025-12-03
> > >
> > > To further substantiate the comparison to ANODE, we made extra experiments to investigate the differences between MODE and ANODE. We find that ANODE cannot easily represent even the toy data (a simplified version of  the branching experiment from Fig 1) since the augmented variables do not receive a direct learning signal during training. In typical ANODE applications, the augmented variable is adapted by a classification loss or pushforward target. This is not the case in our snapshot data and so while ANODE can model crossing (but still deterministic) trajectories, they are not well-adapted to the branch point. ANODE’s learned velocity field was very close to the one obtained with the simpler NODE model. In terms of Wasserstein distance from the ground truth end states, ANODE (resp. NODE, resp. MODE) obtained 0.2318 (resp. 0.2249, resp. 0.1125).  Moreover the augmented variable oscillates around 0 along the trajectories (which means the model doesn’t use it).  MODE, on the other hand, doesn’t need augmented variables; it simply learns to switch between dynamical systems on the fly. Furthermore, while ANODE is an expressive model, MODE has the benefit of being simple and explainable. We view these traits as essential for the scientific settings we have considered, which is why we did not consider replacing the sparse regressors with an ANODE.
> > >
> > > The reference to Figure 17 was as a response to the request for an ablation study as to the effects of the two competing losses (the load-balancing and entropy terms), and shows the balance between them.

---

### Official Review · Reviewer_qMFy · 2025-10-28

**Soundness:** 3
**Presentation:** 3
**Contribution:** 3
**Rating:** 6
**Confidence:** 4

**Summary:**

The paper introduces MODE (Mixture Of Dynamical Experts), a mixture-of-experts framework for snapshot dynamical data that jointly (i) clusters heterogeneous dynamical regimes and (ii) forecasts across regime switches/branching. Each expert is a sparse symbolic regressor, combined by a gating distribution and per-expert isotropic noise. Empirically: (i) on elementary dynamics, MODE yields NMI/ARI ≈ 0.96–1.00, strongly outperforming GMM and spectral clustering and approaching a supervised MLP; (ii) on synthetic forecasting tasks, MODE achieves lower Wasserstein distances than MLP/SINDy and commits to branches; (iii) on U2OS scRNA-seq, a 2-expert MODE matches FUCCI cycle vs. exit AUC = 0.98 over 10 seeds.

**Strengths:**

1. This paper proposes a snapshot-trained MoE for dynamics with interpretable SINDy-style experts, with gating regularizers to avoid collapse and stochastic rollout that commits to fates. Which I believe is new for this field.
2. The method is evaluated on strong and fair elementary benchmarks, which confirms its performance.
3. The paper is well-written, such as the objective (Eq. 3), regularizers (Eq. 4), rollout (Eqs. 5–6) and data generation (appendices) are explicit.

**Weaknesses:**

1. The decomposition between expert field and stochastic term is not probed.
2. Results focus on low-D synthetic (2–3D) and PCA-5 for scRNA. Please add OOD tests.

**Questions:**

1. Please consider adding switching-model baselines (Switched Flow Matching; Neural MJP; mixture-NODEs with gate). Use the same K and similar parameter counts. It would be interesting to see the performance gaps.
2. MODE improves W2 but slightly loses W1,x to MLP (0.1363 vs 0.1284). Please explain the trade-off and add per-axis ablations.

---

> ### Author Response · Authors · 2025-11-21
>
> We thank the reviewer for their kind remarks. Please see below our answers to each of the questions.
>
> - “The decomposition between expert field and stochastic term is not probed.”
> To illustrate the effects of the stochastic switching during rollout, we conducted experiments with mean or winner-take-all rollouts in Appendix Sec. C (“Rollouts”). In general, we find that the stochastic switching scheme more faithfully captures the dynamics of the systems, and are an integral to accurate modeling of the underlying system. We performed a few experiments in which the sigma of Brownian noise was set as the expert sigma discovered by MODE, but saw little difference in the forecasting results. We discuss extensions along these lines in the Discussion.
>
> - “Results focus on low-D synthetic (2–3D) and PCA-5 for scRNA. Please add OOD tests.”
> Our implementation of MODE relies on sparse activations of polynomial basis functions to model the full space. As such, where possible, we utilized dimensionality reduction methods during modeling. We would like to emphasize that for the scRNA data, these PCs were expressive enough to describe the phenomena of interest. To illustrate this, we mapped back from PCA space into gene space, and were able to carry out an analysis into the interactions between genes, showcased in Fig. 6.
>
> - Regarding testing on OOD data, please note that scRNA data is already quite noisy, including positions of cells which are quite unlike those closest to them. In our experiments, we saw that MODE still faithfully captures the different regions of space, even when exposed to such noisy testing data. Truly OOD data could come from ablating whole portions of the cycle, in which case our model like all other dynamical models would probably struggle.
>
>
> - “Please consider adding switching-model baselines (Switched Flow Matching; Neural MJP; mixture-NODEs with gate)”
> We agree with the reviewer that our work will benefit from further comparisons to baselines in the literature of switching mechanics, such as the methods suggested. We would like to emphasize, however, that we focused on snapshot data (i.e. without a time ordering), which makes our research setting rather novel. The papers suggested by the reviewer all use time-indexed samples (i.e. time series), which simply do not exist for omics studies in computational biology, as there are no trajectories that link between cells. There are not even ground truth “before-and-after” labels in our problem setting, which would be relevant to OT or flow matching (including switched flow matching which is just FM plus kmeans).
>
>
> - Instead, we have added comparisons to two additional baselines, HybridSINDy and MODE-NN, which indeed can be trained on snapshots. HybridSINDy (Mangan et al., 2018) is a hybrid approach to modeling dynamical systems, where points are first clustered and then separately fitted. MODE-NN is a variant of MODE using neural networks instead of the sparse dictionary of basis functions in the standard formulation of MODE. Both baselines underperform compared to MODE, as shown in the updated Table 2 and in App. Sec. C (“The role of sparsity”).
>
>
>
> - “Please explain the trade-off and add per-axis ablations.”
> First, we would like to ask for clarification in which types of ablation the reviewer would like to see.
>
> Second, we believe that the advantage of our model particularly in the y marginal is due largely to the way we model the stochastic rollout. The true data commits to its trajectories immediately inside the box, whereas ours vacillate between both until reaching the low-entropy zone outside the switching region. Cells that leave through the front of the switching zone commit too “late” (i.e. too far along the x axis). However, cells that leave through the top and bottom of the switching zone commit on time. We have included new experiments with different types of rollouts in App. Sec. C (“Rollout”).

---

> > ### Comment · Reviewer_qMFy · 2025-11-27
> >
> > I would like to thank authors for the rebuttal. After carefully reading the rebuttal and the revised manuscript, my overall assessment of MODE remains broadly positive, and several of my earlier concerns have been meaningfully, though not completely, addressed.
> >
> > On the baseline side, I had originally asked for switching-model baselines such as Switched Flow Matching, Neural MJPs, or mixture-NODEs with a gate, under comparable K and parameter budgets. The authors correctly point out in the rebuttal that these methods assume time-indexed trajectories, whereas their setting is purely snapshot-based without ground-truth temporal ordering, which indeed makes a direct comparison non-trivial. In the revision they instead add two more relevant snapshot-trained baselines, HybridSINDy and MODE-NN, both of which are clearly described and evaluated under the same data and training pipeline. I still think that at least one time-series switching baseline, even in a simplified synthetic setting where trajectories are available, would have helped position MODE more clearly in the broader literature on mixture dynamics. But given the snapshot-only omics motivation, the added baselines are reasonable and materially improve the fairness and completeness of the empirical evaluation.
> >
> > Regarding dimensionality and OOD behavior, the core experiments are still primarily in low-dimensional synthetic systems (2–3D) and PCA-reduced scRNA-seq (5D). The authors justify PCA-5 by showing that the recovered modes in PC space can be mapped back to gene space and used for interpreting gene interactions, and they add an additional real dataset (human fibroblasts) to demonstrate that the approach is not limited to U2OS. This does help on the “is this biologically meaningful?” axis. However, the paper still does not include explicit, structured OOD tests (e.g., holding out a region of phase space, varying noise levels or sampling density in a controlled way, or training on partial cycles and evaluating on missing phases). The authors argue that real scRNA measurements are already noisy and effectively OOD at the single-cell level, and that MODE remains stable under such noise. While that is plausible, it is more of a qualitative argument than a controlled robustness experiment. I thus still view OOD generalization as a limitation of the current empirical story, though not a fatal one.
> >
> > For such reason, my overall recommendation is unchanged. This is a solid, interesting contribution that sits marginally above the acceptance threshold. I would keep my previous score and confidence.

---

> > > ### Author Response · Authors · 2025-12-03
> > >
> > > We thank the reviewer for clarifying their intentions and suggesting specific experiments.
> > >
> > > To emphasize, MODE is deployed on snapshot data, with which most methods that depend on time-series are incompatible. Nonetheless, we have explored the option of using existing time-series methods, such as [1], on a time-series version of our Goldbeter data. We find that these models are outperformed by MODE, when using the default hyperparameter settings. In future versions of the work, we will compare to better tuned versions. Again, though, we consider these methods as incompatible with the types of data expected in the biological settings which were the focus of our work.
> > >
> > > To better demonstrate the performance of MODE on OOD data, we have reconsidered the Goldbeter oscillator scenario again, where a contiguous sector of the cycle is missing (Appendix B5). At test time, we then try to classify this data into cycling versus exiting. Even though this data isn’t observed during training, we find MODE capably and correctly classifies the OOD data.
> > >
> > > [1]: Linderman, Scott, et al. "Bayesian learning and inference in recurrent switching linear dynamical systems."(2017)

---

### Official Review · Reviewer_fdAi · 2025-10-31

**Soundness:** 3
**Presentation:** 3
**Contribution:** 2
**Rating:** 2
**Confidence:** 4

**Summary:**

The authors present MODE (Mixture of Dynamical Experts), a framework that decomposes complex and noisy biological dynamics into components. MODE enables unsupervised discovery of dynamical regimes and accurate gene expression forecasting across regime transitions. Applied to synthetic and single-cell RNA sequencing data, MODE effectively distinguishes proliferation and differentiation dynamics and predicts cell fate commitment.

**Strengths:**

1.	The manuscript is clearly written and well organized.
2.	The experiments cover both simulated and real-world datasets, providing comprehensive validation.

**Weaknesses:**

1.	The advantage or the main goal of developing MODE is not that meaningful. There are many RNA velocity models, like VeloVAE (Gu et al, ICML 2022, LatentVelo ), that are explicitly designed to model cell lineage bifurcation. This significantly reduces the novelty and practical impact of this study.
2.	The result from the benchmarking is not convincing, since GMM and spectral methods are quite simple and may not be suitable for complicated data. The authors should compare the performance with other methods mentioned in the related work, like flow matching based methods (Meta Flow Matching) and maybe some RNA velocity models like scVelo (Bergen et al, 2020). And it will also be helpful to compare with other MoE models, like DynMoE (Guo et al, 2025).
3.	The scalability is questionable, which is crucial for real single cell RNA sequencing data. The U2OS cell line dataset contains only about 1,000 cells, which is a very limited number in reality. It will be helpful if the authors could run the method on a larger dataset, like the mammalian organogenesis dataset (Cao et al, 2019), which also has branching differential trajectories for this dataset. And I also suggest including metric like cross-boundary direction correctness (CBDir, Qiao et al, 2021) for the real dataset.

**Questions:**

1.	Why didn't you just use traditional precision, recall and F1 metrics for evaluation?
2.	How much compute time does the MODE model need in each of your experiments?
3.	When generating the data, how will different noise levels affect the model performance?

---

> ### Author Response · Authors · 2025-11-21
>
> We thank the reviewer for these critical remarks, which have prompted us to make some significant changes to the language in the paper. We agree that we should have been clearer about our approach’s relation to other ideas from RNA velocity literature, which we take to be fundamentally separate approaches.
>
>
> - The works mentioned by the reviewer, like VeloVAE and LatentVelo, are primarily concerned with the estimation of the velocities of genes, based on the interactions between their spliced and unspliced counts, as well as pseudotimes for each cell in a differentiation process. For bifurcating data, these methods rely on label-specific transcription rates,  which in essence assumes that cells have decided their fate right at the start of their trajectory. This point, that cell trajectories are deterministic, is even mentioned as a weakness in the Discussion of LatentVelo. This is further exacerbated in cases such as the cell-cycle exit in Figures 5, Figure 15, which are typically not labeled.  Moreover, both methods were not designed to work with cycling data coupled with branching points.
>
>
> - We view the velocities estimated by the methods above as a first-step - the input - to our work. Building on these velocities, MODE is used to build a population-level model for the governing equations of the process, as opposed to the local transcription dynamics of the former. Taking this view, we can model the full population dynamics and take into account the explicitly stochastic nature of the dynamics in biological systems. We believe this is both computationally and biologically different, and we emphasize this in the revised manuscript.
>
>
> To address the individual remarks:
>
>
> - “The advantage or the main goal of developing MODE is not that meaningful”
>  We agree with the reviewer that our original description left the comparison to these other works somewhat unclear. In a revised Introduction and Related Works section, we emphasize that our approach is not to simply model lineages and branching, but rather to learn dynamics which cannot mathematically be represented as a single flow. This is clearest in the cell cycle data, where perpetually cycling populations overlap with intermittently exiting ones, including in the new experiment on fibroblast (Fig. 15).
>
>
> - “It will be helpful if the authors could run the method on a larger dataset”
> We have added a new experiment on human fibroblasts (Fig. 15). This new dataset has 5,367 cells, with dynamics that are similar to the U2OS data. On this data, as in the U2OS data, MODE is able to clearly delineate between the cell-cycling regime and the exit phenotype.
>
>
> - We agree that looking at even larger datasets, such as the mammalian organogenesis dataset, would be of great interest. However, in this work, we were primarily interested in systems which branch into two distinct behaviors, where stochastic assignment to different experts is crucial to accurately capture the underlying dynamics. As such, we view the exploration of more complex datasets, such as the one suggested, as out of scope for this work and of great interest as future applications of our work. However, given the favorable scaling in the cell dimension that our model exhibits (see remarks on runtime), we do not see this as an infeasible next step.
>
>
> - “Why didn't you just use traditional precision, recall and F1 metrics for evaluation?”
> We agree that these metrics are clearer and now Table 1 reproduces our results in this manner.
>
>
> - “How much compute time does the MODE model need in each of your experiments?”
> As we note in App. Sec. C2, full training times run from about a minute for the clustering experiments to about ten minutes for the FUCCI experiment (one seed). For this latter experiment, full training on a Macbook Air took about 53 seconds. It’s true that run times are a concern in omics data, which can be massive, but we believe our approach is actually very suited to this scale. Indeed, our use of snapshots makes MODE simulation-free, which speeds up dynamical modeling significantly. Even with 100x the number of cells, training time would be reasonable. Scaling to higher gene counts is also interesting, but our current results show that even learning on PCA allows us to map back to meaningful biological mechanisms.
>
>
> - “When generating the data, how will different noise levels affect the model performance?”
> We were also interested in this question as our ultimate aim is to apply the method on real, noisy data. Our benchmarking for noise on the clustering data is found in  Appendix B.2. While running systematic studies on forecasting in the presence of variable noise levels is beyond our current study, we do believe that our two real-data experiments show that MODE is applicable to realistic noise settings.

---

### Official Review · Reviewer_RZaS · 2025-11-01

**Soundness:** 2
**Presentation:** 3
**Contribution:** 2
**Rating:** 4
**Confidence:** 4

**Summary:**

The paper introduces MODE (Mixture of Dynamical Experts), a new framework designed to model complex systems, like those in computational biology (e.g., cell differentiation) where the behavior shifts dramatically over time or across different states. Unlike traditional models (like Neural ODEs) that assume a single, smooth governing rule, MODE uses a mixture-of-experts approach. It learns multiple simple, interpretable equations (i.e., experts) and a gating function that dynamically chooses the right equation for a given state. This allows MODE to successfully model systems that bifurcate or transition between different operating regimes.

**Strengths:**

- MODE’s formulation as a mixture of sparse dynamical regressors with neural gating is conceptually simple yet powerful. It bridges classical sparse regression (e.g., SINDy) and modern mixture models, resulting in interpretable and flexible dynamics decomposition.

- Challenges introduced in L37-43 with an interesting example of RNA sequencing make sense. Also, the problem statement (Consequently, a large body of research at the intersection of computational biology and data-driven dynamical systems has been devoted to the modeling of snapshot data. Regard the action error diagnosis as a node classification task.) is reasonable.

- The model’s ability to discover latent regimes without supervision is well demonstrated, highlighting its potential for uncovering hidden cellular states or transitions from high-dimensional, noisy biological measurements.

- The evaluation spans from controlled synthetic systems (bistable, predator–prey, Lorenz) to biological switching processes and real single-cell data, which demonstrates the model’s versatility and robustness.

**Weaknesses:**

- Figure interpretation and clarity. It is unclear whether the x-axis in Figure 1 represents time. The visualization appears to suggest that blue cells evolve into red and green cells over time. In the overlapped region (left panel), do the red and green cells physically interact, or does their spatial overlap simply obscure individual dynamics? I would appreciate further clarification on how overlapping dynamical regimes introduce modeling challenges and whether this overlap is a visualization artifact or a true physical mixture.

- Is the research problem novel? The general problem of learning systems governed by multiple, switching, and partially unknown dynamics has been extensively studied in the literature (e.g., Graph Switching Dynamical Systems, ICML 2023). It is not fully clear what unique challenge this paper addresses beyond existing frameworks for switching or hybrid dynamical systems.

- Is MODE technically novel? The MODE framework appears conceptually similar to standard mixture-of-experts models with sparse MAP estimation. The objective in Equation (3) essentially corresponds to a weighted MAP formulation, and the comparison baselines (GMM, supervised MLP, NODE) are arguably too weak or mismatched for a fair assessment. It would strengthen the paper to demonstrate MODE’s advantage over stronger baselines explicitly designed for switching or compositional dynamics.

- Motivational gap regarding branching systems. The paper motivates MODE by emphasizing that biological systems may bifurcate into multiple branches. However, it is not clear why existing models could not be independently trained for each branch (e.g., fitting separate NODEs for red and green trajectories). If MODE’s advantage lies in discovering branches without supervision, this distinction should be emphasized more clearly.

- How do we know how many experts will be required? The number of experts (K) seems to correspond to the number of distinct dynamical regimes, which in practice may require prior domain knowledge. It remains unclear how MODE performs when K is misspecified or when the true number of regimes is unknown, which could be an important consideration for real-world biological applications.

- In Line 185, the authors assume that each sample is governed by one expert, implying no interaction across regimes. If that assumption holds, why not simply learn from data after regime transitions are completed, rather than during overlap? Clarifying this design choice would help justify the need for mixture modeling during ambiguous transitions.

- Equation (4) introduces pi_s(x) as the expert assignment distribution, but it is unclear how this distribution behaves in practice. Are the mixture weights highly non-uniform across regimes, and how sensitive are results to the entropy or balancing regularizers?

**Questions:**

Please see the weaknesses.

---

> ### Author Response · Authors · 2025-11-21
>
> We thank the reviewer for the probing remarks, which have prompted several important changes. Our Related Works and Methods sections now argue for our method’s computational and biological novelty. We argue that switching/hybrid dynamical systems approaches either rely on trajectories (not unordered snapshots) or pre-clustering of dynamical regimes (e.g. Hybrid SINDy, Mangan et al., 2018). MODE separates itself from these methods by its applicability to snapshots and its joint clustering/model discovery approach. We demonstrate this with the inclusion of two new baselines and a hyperparameter study emphasizing the crucial role of sparsity in our framework. In detail:
>
> - “Figure interpretation and clarity….” In Fig.1, the spatial overlap implies that cells co-mingle in gene expression space but have different dynamics: green cells trend up with north-eastern facing vectors and red trend down with south-eastern. This can happen when the dynamics change from extra-transcriptomic (e.g. proteomic) factors not measured in scRNAseq. In short, this is not a visual artifact but a true mixture which is prevalent in scRNAseq data, especially in cell cycle data. The MS text has been adjusted to reflect this.
>
>
> -  “Is the research problem novel?” We believe our focus on snapshot data (i.e. without a time ordering) makes our research setting rather novel. For example, the ICML 2023 paper cited by the reviewer uses time-indexed samples, as do Ojeda et al., 2021 and Seifner and Sanchez, 2023. Snapshot data are the standard for omics studies, however. Further, from a biological point of view, to our knowledge the application of any switching models at all (snapshot or otherwise) is wholly novel in omics.
>
>
> -  “Is MODE technically novel?” We have adjusted the text and added new experiments to emphasize the technical novelty of our approach. We argue that the joint learning of dynamical regressors and expert allocations is a technical novelty, and it leads to the conclusion that a sparsity criterion on the learned dynamics can have a strong interaction with mixture modeling: roughly, the sparser the regressors, the better the clusters. We now compare MODE to a related mixture framework, Hybrid SINDy (Mangan et al., 2018) whose expert allocations are instead learned before dynamical regressors. Hybrid SINDy struggles to identify the true dynamical regimes (Fig. 10b) and performs significantly worse in forecasting (Tab. 2, row 3). This shows that co-learning the gating network and dynamics,our approach, is fundamentally different from pre-clustering. New experiments (Figs. 9,11) also show that adjusting sparsity in MODE systematically affects discovered regimes. This result is bolstered with the new inclusion of an ablated model, MODE-NN which lacks sparsity (Table 2, row 4).
>
>
> - “Motivational gap regarding branching systems”  We now emphasize in the manuscript that the unsupervised discovery of dynamical regimes is one of the model’s strong suits and it can be intuitively controlled with sparsity (App. Sec. C, “The Role of Sparsity”).
>
>  - “How do we know how many experts will be required?” It is true that the number of experts, K, is often not known in advance. We now detail an algorithm that automatically adjusts the number of experts during training based on the Akaike Information Criterion. This is a standard criterion in model discovery and it works seamlessly with our approach. We now include a new experiment (App. Sec. C “Adaptive Expert Selection”) which shows how this algorithm then consistently converges to K=2 experts on the FUCCI data across 15 reruns.
>
>
>  - “ why not simply learn from data after regime transitions are completed” We would like to emphasize that in real data, we would expect the regime of the switching to be unlabeled. In our experiments, such labels were used exclusively for validation, and not utilized during training. For instance, the true regimes in the FUCCI data are not easy to discover without prior knowledge, and yet our framework can consistently find them. In contrast, our new experiment with Hybrid SINDy (Table 2, Fig. 10b), shows that discovering these regimes independently from model fitting can be tricky, even on relatively simple data. In short, we believe that jointly learning dynamical models and expert allocations (i.e. MODE) is a more straightforward approach.
>
>
> - “it is unclear how this distribution behaves in practice” There are a few examples of the gating distribution in Fig 5b, 8 and 9. We would like to clarify what the reviewer means about the weights being non-uniform across regimes. In general, we find that distributions shift abruptly between the true regimes. We now also include experiments in which sparsity (Fig. 11) and entropy regularizers (Fig. 15) are systematically varied, showing how the former tends to stabilize expert assignments in training and the latter can be comfortably optimized in a grid search.

---

### Official Review · Reviewer_6uqC · 2025-11-06

**Soundness:** 3
**Presentation:** 3
**Contribution:** 2
**Rating:** 6
**Confidence:** 3

**Summary:**

The paper introduces a new method called MODE (Mixture of Dynamical Experts) to address the challenge of modeling complex systems, particularly in computational biology. It focuses on the difficulties posed by sparse and unordered `snapshot’ data that often exhibit multiple overlapping behavioral regimes and branching dynamics. MODE tackles this gap by using a gated mixture model that decomposes complex, ambiguous flows into distinct dynamical components. The authors demonstrated that MODE outperforms existing approaches in unsupervised classification of dynamical populations under noise and limited sample sizes, using both synthetic and real-world single-cell RNA sequencing data.

**Strengths:**

1) The paper abstract, introduction and related work are written very well and motivate their method.
2) They discussed and prioritized method interpretability which is crucial for scientific applications.
3) They applied the method to both real world and synthetic data.
4) They explicitly explained the distributions and the assumptions about (most) variables.

**Weaknesses:**

1) The method itself should be more detailed, i.e. you state the model, but the fitting procedure / algorithm could be more clearly explained.
2) Model-wise, from what I understand, it seems like the authors had an implicit assumption that each snapshot reflects a single state (i.e., comes from one most likely expert). I would assume that during its cycle, a cell's evolution may be governed by multiple experts simultaneously, e.g., experts that capture division-related signals as well as growth signals that may reflect different dynamics and can be combined for a more flexible expert.
3) While I recognize that this paper's main goal is to describe the method, I miss a short discussion about what the method tells us about the biology (rather than only e.g. predictions). E.g. How much in advance in time can you predict future differentiation?
4) In the related work, there is a missing discussion on decomposed dynamical system models [1,2] and their extension to long term forecasting [3].
4) With respect to figure 4, you talk about SINDy but I cannot see it presented.
5) The method assumes that the basis functions that define the dynamics (Z) are known and are polynomials. It is built on setting these a priori, which requires choosing/knowing the appropriate polynomial basis.
6) Can you clarify what is B? (unless I missed it, I cannot find where it is defined)
7) Small typo: title of section 3 should be method, not methods.


[1] Mudrik, N., et al. (2024). Decomposed linear dynamical systems (dlds) for learning the latent components of neural dynamics. JMLR

[2] Chen, Y., et al. (2024). Probabilistic decomposed linear dynamical systems for robust discovery of latent neural dynamics. NeurIPS

[3] Mudrik, N., et al. (2024). LINOCS: Lookahead Inference of Networked Operators for Continuous Stability. TMLR

**Questions:**

1) How would you model a case where each snapshot evolves by the rules of multiple co-active experts that govern its dynamics?
2) Did you try to look at the 2nd or 3rd most likely expert in every time point? Maybe you will reveal differences within the apparently the same cell state across snapshots that can further reveal their fate even earlier in time?
3) How much does the degree of the polynomials affect the result? Is there a way to fit / infer the basis Z rather than setting it as a hyperparameter?
4) What can the method tell us about the biology / biological processes in future datasets that existing methods cannot?

---

> ### Author Response · Authors · 2025-11-21
>
> We thank the reviewer for the constructive remarks, particularly as they touched on our approach’s applications to biology. In response to these remarks, we have made several important clarifications in the text as well as new experiments to highlight our method’s biological interpretability. We enumerate the changes to the new MS version here:
>
>
> - “The method itself should be more detailed…” Appendix C now has a more detailed description of both the model and especially the fitting procedure and algorithm.
>
>
> - “ it seems like the authors had an implicit assumption that each snapshot reflects a single state” The reviewer raises an important point, and indeed cellular dynamics can be governed by multiple co-existing processes. In our setting, these can be modeled as dynamics with either soft (i.e. dynamics are weighted by multiple experts simultaneously) or winner-take-all (i.e. follow most likely expert) guidance by the experts. MODE can already take steps towards modeling these co-existing regulatory processes, as we now show in a new experiment in App. Sec. C (“Rollouts”). We report results for each of these rollout types in the case of the Goldbeter oscillator. This allows our framework to model exactly the types of co-active experts the reviewer suggests.
>
>
> - “ I miss a short discussion about what the method tells us about the biology…” We agree, and we have therefore added new results (Sec. 3.4, Fig. 6) in which MODE is used to infer gene regulatory changes during cell cycle exit. Unlike other methods, we find that MODE is capable of inferring dynamic changes in regulatory network architectures which result from categorical shifts in transcriptional behavior (e.g. cell cycle exit). MODE is built of interpretable building blocks, which can be probed after fitting, so that we can now query precise gene-gene interactions which shift during cycle exit (Fig. 6b). We confirmed that the identified interactions are indeed relevant to cell cycle exit, supporting MODE's use specifically for studying discovering transcriptional dynamics, including In future biological settings where transcriptional drivers of categorical shifts are unknown.
>
>
> - “The method assumes that the basis functions that define the dynamics (Z) are known…” We would like to clarify that, while our use of SINDy does require positing some basis, our method does not require detailed prior knowledge of the true basis functions. Indeed, the Goldbeter oscillator (Eqs. 13-15) is governed by rational functions which lie outside our polynomial basis, but MODE’s polynomial basis still expresses the underlying dynamics quite well. We also now examine the relation between forecasting performance and the maximum polynomial degree in a new experiment (App. Sec. C Fig. 16). Our choice to use symbolic regression is guided by the following criteria: (1) in the case that the basis does match the true system, the discovered equations will be directly interpretable, (2) the use of a finite basis allows us to use sparsity regularization, which, as we demonstrate in a new experiment (App. Sec. B, “The Role of Sparsity”), aids in the discovery of boundaries between dynamical regimes, and (3) blackbox regressors like neural ODEs perform worse, as we also demonstrate with a new baseline (MODE-NN in Tables 2-3).
>
>
> Other small changes:
> - Related work now contains the reviewer’s citation suggestions.
> - SINDy as a sub-method is now described in more detail in App. Sec. C.
> - Typographical (particularly B => N) and stylistic corrections have been made following reviewer suggestions.

---

### Author Response · Authors · 2025-11-21

We thank all five reviewers for their constructive remarks. We have extensively revised the manuscript, whose new version has been uploaded. The main changes are

- We have clarified the problem setting and the technical novelty of our approach: joint learning of dynamics and expert allocation on snapshot data. This is demonstrated through several new studies on the role of sparsity in our model and comparison to other switching/hybrid models which lack this joint learning approach.

- We have demonstrated that the number of experts can be discovered reliably from data in the context of our scRNAseq experiments (Fig. 18). It does not need to be known in advance.

- We have added extensive new biological analysis in our scRNAseq experiments (Fig. 6), showing that MODE can be used to map back to raw gene space to discover the molecular mechanisms driving cell cycle exit.

- We have added a new experiment on modeling cell cycle in human fibroblasts (Fig. 15), in which our approach performs favorably.

- Various hyperparameter and ablation studies have been added concerning all the main components of our proposed framework (Figs., 9, 10, 11, 16, 17).

---

> ### Author Response · Authors · 2025-12-03
>
> We  thank  the  reviewers  for  their  constructive comments and overall positive evaluation of our manuscript, which they found “outperform[ed] existing approaches in unsupervised classification of dynamical populations under noise and limited sample sizes, using both synthetic and real-world single-cell RNA sequencing data” while being “written very well”  and “prioritiz[ing] method interpretability which is crucial for scientific applications” (Reviewer 6uqC). They describe the method as “simple yet powerful … resulting in interpretable and flexible dynamics decomposition” with “[the] model’s ability to discover latent regimes without supervision … highlighting its potential for uncovering hidden cellular states or transitions from high-dimensional, noisy biological measurements” on “real single-cell data, which demonstrates the model’s versatility and robustness” (Reviewer RZaS). Reviewer qMFy considers our work “new for this field”, while being “evaluated on strong and fair elementary benchmarks, which confirms its performance” (Reviewer qMFy). Our work “clearly identifies the failure of traditional flow-based models” and provides “a unified perspective to address key limitations of existing methods”, in particular “the inability of flow-based models to handle branching dynamics, the computational complexity of switching system inference, and the lack of interpretability in many neural approaches”, with a “framework [that] is great for its simplicity … which is a significant advantage for scientific applications” (Reviewer 1SNf).
>
> We have addressed all comments and suggestions raised by the reviewers, in a point-by-point response to each reviewer, and have revised our manuscript and appendix in a manner which we believe further strengthens our work. Following up from subsequent discussions with reviewers, we added still further changes to the manuscript so that each reviewer request was met. The main points of the revision are:
>
> We have clarified the problem setting and the technical novelty of our approach as recommended by Reviewers 6uqC, RZaS, and fdAi.
>
>
> As suggested by Reviewers qMFy and 1SNf, we have added additional baselines which further demonstrate that MODE outperforms other methods, while remaining wholly interpretable.
>
>
> Importantly, we have also considered a setting where the true number of experts, K, is unknown in advance, resolving one of the main critiques of Reviewers 6uqC, RZaS, and 1SNf. Figure 5 in the main text has been updated to show that it is possible to estimate this value in parallel with the dynamics themselves.
>
>
> Following the suggestion of Reviewer 6uqC, we have extended the discussion and added new results (new Sec 2.3) demonstrating how MODE can be used to predict changes in gene regulatory networks accompanying the exit from cell cycle, directly leveraging MODE’s interpretability to make novel biological predictions.
>
>
> We have extended the discussion in the Related Works to clearly differentiate our work from recent advances in computational biology.
>
>
> Following suggestions from Reviewers 6uqC and 1SNf, we have added extensive ablation studies, probing the effects of both the size and degrees part of the polynomial dictionary in Fig. 17 of the Appendix.

---

### Comment · Area_Chair_Gwnt · 2025-11-26
**Reminder to Engage!**

Dear Reviewers,

We are one week away from the end of the discussion period and the review responses have been posted. If you have not done so already, please read the response and check if the authors have addressed your concerns. Also please acknowledge the review by responding and stating how the response (and updated manuscript if provided) does or does not change your evaluation of the work. Earlier responses allow for meaningful engagement and potential for further clarification.

-Area Chair

---

### Meta-Review · Area_Chair_EfVq · 2025-12-20

**Summary:**

The paper proposes MODE, a framework for learning dynamical systems from snapshot data without temporal ordering. The method combines a neural gating network with sparse symbolic regressors to decompose complex dynamics into components, specifically targeting branching and switching behaviors often found in biological systems like single-cell RNA sequencing data.

While the reviewers acknowledged the clear motivation, the methodological contribution is best viewed as a standard assembly of existing components (MoE, SINDy, gating) rather than a significant advancement. A major criticism is that the evaluation does not sufficiently demonstrate the scalability to high-dimensional genomics to justify its empirical significance. Given these concerns regarding incremental novelty and limited empirical breadth, the recommendation is for rejection.

**Reviewer Concerns:**

The authors' rebuttal addressed the concern regarding the manual selection of experts by introducing an AIC-based adaptive strategy. The inclusion of Hybrid SINDy and ANODE baselines added context for the method's performance in relative terms. Outstanding concerns include technical novelty, scalability (the added fibroblast dataset is a step forward but still falls short of the scale requested) and partially convincing OOD tests.

**Reviewer Scores:**

Reviewer fdAi and RZaS would likely not increase their score as their primary concerns regarding scalability and ML significance remain valid. Reviewer 1SNf might be receptive to raising score due to the addition of adaptive expert selection and ANODE comparison but some skepticism still remains. Reviewer qMFy would likely maintain their score. While generally positive, the OOD testing and the low-dimensional nature of the experiments were acknowledged, solidifying their view of the paper as a niche contribution rather than a clear accept.

---

### Decision · Program_Chairs · 2026-01-26

Reject